# Contrastive Graph Structure Learning via Information Bottleneck for Recommendation

**Chunyu Wei**[1]*,**Jian Liang**[1]*, **Di Liu**[1], **Fei Wang**[2]
[1]Alibaba Group, China
[2]Department of Population Health Sciences, Weill Cornell Medicine, USA
weicy15@icloud.com
{xuelang.lj, wendi.ld}@alibaba-inc.com
few2001@med.cornell.edu

## Abstract

Graph convolution networks (GCNs) for recommendations have emerged as an important research topic due to their ability to exploit higher-order neighbors. Despite their success, most of them suffer from the popularity bias brought by a small number of active users and popular items. Also, a real-world user-item bipartite graph contains many noisy interactions, which may hamper the sensitive GCNs. Graph contrastive learning show promising performance for solving the above challenges in recommender systems. Most existing works typically perform graph augmentation to create multiple views of the original graph by randomly dropping edges/nodes or relying on predefined rules, and these augmented views always serve as an auxiliary task by maximizing their correspondence. However, we argue that the graph structures generated from these vanilla approaches may be suboptimal, and maximizing their correspondence will force the representation to capture information irrelevant for the recommendation task. Here, we propose a Contrastive Graph Structure Learning via Information Bottleneck (CGI) for recommendation, which adaptively learns whether to drop an edge or node to obtain optimized graph structures in an end-to-end manner. Moreover, we innovatively introduce the Information Bottleneck into the contrastive learning process to avoid capturing irrelevant information among different views and help enrich the final representation for recommendation. Extensive experiments on public datasets are provided to show that our model significantly outperforms strong baselines. [2]

## 1 Introduction

Recommender systems have been widely deployed to alleviate information overload in diverse scenarios including e-commerce, online news and multimedia contents, which requires high-quality user and item representations learned from the historical interactions [7, 14, 43]. Recently, thanks to the powerful capability in modeling graph-structured data, Graph Convolution Networks (GCNs) provide an efficient way to integrate multi-hop neighbors into node representation learning and show prominent performance in recommendation [37, 30, 8].

Although encouraging performance has been achieved, we argue that most GCN-based recommender models suffer from the following two limitations, of which the impacts on the user's exhibited preference are presented in Fig. 1. i) **Popularity Bias.** Items inherently have different customer sizes, and this imbalance can potentially lead to popularity bias [45]. In most recommender systems, the customer size for items usually follows a long-tail distribution, which means a few items have massive customers while the majority have few customers. Similarly, most users have few interactions. This skewed data distribution will bias GCN-based models towards the popular users and items easily

---

*Equal contributions from both authors. This work is done when Chunyu Wei works as an intern at Alibaba.
[2]The code is available on https://github.com/weicy15/CGI.

during multi-hop convolution, which may hamper the representation learning. ii) **Interaction Noises.** User-item interactions usually contain noises especially in the scenarios with only implicit feedbacks (e.g., clicks and purchases). More specifically, these noisy edges in the bipartite graph are not necessarily aligned with user preferences [18], since it's common that the user clicks something by mistake or finds something boring after purchasing. GCN-based models are known to be vulnerable to the quality of the input graphs [44], which means aggregating misleading neighborhood information is likely to lead to sub-optimal performance. Recent advances in graph contrastive learning [27, 38] have identified an effective training scheme for mitigating popularity bias and increasing robustness for noise on graph-based tasks, which inspire many studies [31, 41, 33] to introduce this training scheme to enhance representation learning for recommendations.

Nevertheless, existing studies have two limitations. First, most methods perform data augmentation by randomly dropping edges/nodes to change the graph structure [31], shuffling the embeddings to corrupt the node representations [41], or relying on pre-defined rules [6]. However, within unsupervised settings, structures created from these vanilla approaches may be suboptimal for recommendation tasks and also lack persuasive rationales for why the randomly dropped edges/nodes alleviate the popularity bias and interaction noises. Like the obtained representation $No.1$ in Fig. 1, structures created from these vanilla approaches may deviate from the

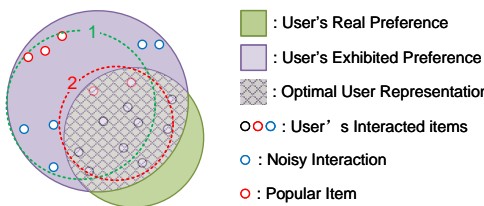

Figure 1: A possible illustration of some user's interactions and preference. Dotted circles denote possible augmentation representations.

optimal area. Second, most methods generate multiple views only to serve as an auxiliary task by maximizing the agreement of node representations among these views, which may force the user or item representation in different views to capture the information irrelevant for the recommendation task. For example, the obtained representation $No.1$ in Fig. 1 contains much information irrelevant to the real preference. So we believe that a good augmentation (e.g., $No.2$ in Fig. 1) should cover as much optimal area as possible while being as small as possible to reduce useless information.

To address the aforementioned limitations, we propose Contrastive Graph Structure Learning via Information Bottleneck (CGI) for recommendation, which contains two key components: **learnable graph augmentation** and **information bottleneck contrastive learning**.

First, we propose learnable graph augmentation to learn whether to drop an edge or node to transform the original bipartite graph into correlated views, which will be jointly optimized with the downstream recommendation in an end-to-end fashion. As a result, these generated views can intentionally reduce the influence of popular nodes while preserving information of the isolated nodes, and thus help to mitigate the popularity bias. The intuition behind is that random dropout will indiscriminately drop nodes or edges regardless of the corresponding node degrees, while by message passing mechanism, GCNs are easier to reconstruct the missing information of popular users or items, but much harder to reconstruct those isolated nodes with few connections, thus may overemphasize those high-degree nodes. These generated views with debiased information are all fed into the GCN-based recommender for multi-view representation learning to increase the ability against popularity bias.

Second, we proposed to integrate different views into a compact representation for the downstream recommendation tasks, which can further improve the robustness of the model. Generally, when information from different views complements each other, it can be expected that the multi-view representation learning approaches can improve downstream performance [28]. So we argue that simply maximizing the mutual information in the conventional graph contrastive learning may push the representations of different views to capture information irrelevant to the downstream task. Inspired by the recent advances of Information Bottleneck (IB) [32], which encourages the representation to capture the *minimum sufficient* information for the downstream task, we utilize the IB principle to minimize the mutual information between the original graph and the generated views while maintaining the downstream recommendation performance of each view. By doing so, the learnable graph augmenters can learn to remove noisy interactions in the original graph as much as possible, since these interactions are of no help for the downstream recommendation. Also, the IB principle helps representations of different views to capture collaborative information of different semantics complement to each other.

The contributions of this paper are summarized as follows. (1) We propose the CGI to construct optimized graph structures by dropping nodes and edges adaptively for the multi-view representation learning of users and items, which provides rationales for alleviating the popularity bias. (2) To efficiently drop information irrelevant to the downstream recommendation, we innovatively integrate information bottleneck into the multi-view contrastive learning process for recommendation and prove that it can better mitigate interaction noises. (3) Experimental results show that our method outperforms the state-of-the-art methods on three benchmark datasets from different domains.

## 2 Related Work

**Graph-based Recommendation** Early works exploiting the user-item bipartite graph for recommendation like ItemRank [3] usually followed the label propagation mechanism to propagate users' preference over the graph, i.e., encouraging connected nodes to have similar labels. In recent years, Graph Convolution Networks (GCNs) have made great progress in representation learning tasks including node classification and link prediction [5, 12, 35]. Motivated by the strength of GCNs, several works [24, 8, 37, 30] have adapted GCNs on the user-item bipartite graph to learn more robust latent representations for users and items in recommender systems.

**Contrastive Learning** Contrastive Learning (CL) [22, 9] was firstly proposed to train CNNs for image representation learning. Graph Contrastive Learning (GCL) applies the idea of CL on GNNs. DGI [27] and InfoGraph [19] learn node representations according to the mutual information between nodes and the whole graph. Peng et al. [15] developed an unsupervised learning model trained by maximizing mutual information of nodes between the input and output of a graph neural encoder. Hu et al. [10] extend the idea to build contrastive pairs between nodes and subgraphs. In addition, GCC [16] designs the pre-training task as subgraph instance discrimination in and across networks and leverage CL to empower GNNs. And a very recent work SGL [31] supplements the classical supervised task of recommendation with an auxiliary graph CL task, which generates multiple views of a node and maximizes the agreement between different views. However, it differs from our work in: (1) SGL [31] generates contrastive pairs by randomly dropping edges/nodes, while our work adopts a learnable augmenter to optimize the generated views. (2) SGL [31] utilizes conventional CL as an auxiliary task by maximizing the agreement of augmentation views, while we propose to encourage the differences between the augmentation views and the original graph.

**Learning by Information-Bottleneck** Information Bottleneck (IB) [23] is an approach based on information theory, which states that if the obtained representation discards information from the input which is not useful for a given task, it will increase robustness for the downstream tasks. Besides, the information bottleneck principle is used in multi-view representation learning [34, 29, 2]. Formally, given the original data $\mathbf{X}$ with label $\mathbf{Y}$, IB is to obtain a compact and effective representation $\mathbf{Z}$ of $\mathbf{X}$. And the objective of the IB principle is as follows:

$$\max_{\mathbf{Z}} I(\mathbf{Y}; \mathbf{Z}) - \beta I(\mathbf{X}; \mathbf{Z}), \tag{1}$$

where $\beta$ is the coefficient to balance the mutual information $I(\mathbf{Y}, \mathbf{Z})$ and $I(\mathbf{X}, \mathbf{Z})$.

Recently, some works proposed to integrate the IB principle into the graph learning process. You et al. [39] propose a variational graph auto-encoder to generate contrastive views and the downstream contrastive learning utilizes IB performing on graph representations as the unsupervised loss. Both Yu et al. [40] and Yu et al. [42] aim to directly reveal the vital substructure in the subgraph level, among which [1] learns a node assignment matrix to extract the subgraph, and implements the IB of two graphs by estimating the KL-divergence from graph latent representation with a statistic network (DONSKER-VARADHAN Representation of KL-divergence). And Yu et al. [42] employ noise injection to manipulate the graph, and customizes the Gaussian prior for each input graph and the injected noise, so as to implement the IB of two graphs with a tractable variational upper bound. Our CGI differs from them, since we do not directly aim to find an optimal graph structure, instead we try to learn the graph structure complementing the original one. Then by integrating different views into a compact representation, we obtain the optimal node representation for the downstream task. Sun et al. [20] learn to mask node feature and generates new structure with the masked feature. Afterward, [20] adopt GNN to learn the distribution of graph representation and utilize the KL-divergence between the learned distribution and the prior distribution to implement the IB.

All these methods aim to find a better structure or representation to replace the original graph for the downstream task, while our CGI follows a multi-view representation learning schema. IB is

utilized to minimize the mutual information between the original graph and the generated views while maintaining the downstream recommendation performance of each view. Besides the noise-invariance property, IB helps representations of different views to capture collaborative information of different semantics that complement each other.

AD-GCL [21] shares some ideas with our CGI but there are fundamental differences. Specifically, AD-GCL focuses on training self-supervised GNNs for graph-level tasks. In contrast, CGI aims to mitigate the popularity bias and interaction noises of node-level collaborative filtering (CF). In addition, AD-GCL adopts an adversarial strategy aiming to maximize the agreement of final representations of different views. Instead, our CGI minimizes the mutual information of different views to capture collaborative information of different semantics. To the best of our knowledge, this is the first study on leveraging the IB principle to enhance graph-based recommendations.

## 3 Preliminaries

**Problem Definition.** Let $\mathcal{U} = \{u_1, u_2, \ldots, u_m\}$ denotes the set of users, and let $\mathcal{I} = \{i_1, i_2, \ldots, i_n\}$ denotes the set of items. We typically use a binary matrix $\mathbf{R} \in \mathbb{R}^{m \times n}$ to store user-item interactions (e.g., purchases and clicks), where $r_{ui} = 1$ indicates that user $u$ consumed item $i$ while $r_{ui} = 0$ means that item $i$ is unexposed to user $u$ or user $u$ is not interested in item $i$. Following most existing works [30, 8], we represent interaction data as a user-item bipartite graph $\mathcal{G} = \{\mathcal{V}, \mathcal{E}\}$, where the node set $\mathcal{V} = \mathcal{U} \cup \mathcal{I}$ and the edge set $\mathcal{E} = \{e_{ui} | r_{ui} = 1, u \in \mathcal{U}, i \in \mathcal{I}\}$. The adjacency matrix $\mathbf{A}_{\mathcal{G}}$ can be formulated as follows:

$$\mathbf{A}_{\mathcal{G}} = \begin{bmatrix} 0 & \mathbf{R} \\ \mathbf{R}^T & 0 \end{bmatrix}. \tag{2}$$

With respect to the adjacency matrix $\mathbf{A}_{\mathcal{G}}$, the degree matrix $\mathbf{D}_{\mathcal{G}} \in \mathbb{N}^{(m+n) \times (m+n)}$ is a diagonal matrix, in which each entry $\mathbf{D}_{\mathcal{G}}[i, i]$ denotes the number of nonzero entries in the $i$-th row of $\mathbf{A}_{\mathcal{G}}$.

**GCN Paradigm.** The core of graph convolution on graph $\mathcal{G}$ is to update the ego node by aggregating the representations of its neighbor nodes, which can be formulated as follows:

$$\mathbf{E}^{(l)} = GCN(\mathbf{E}^{(l-1)}, \mathcal{G}), \tag{3}$$

where $\mathbf{E}^{(l-1)}$ is the current representations of nodes and $\mathbf{E}^{(l)}$ is the updated representations after the graph convolution layer. $\mathbf{E}^{(0)}$ is the initial inputs, which are usually the ID embeddings (trainable parameters). From the vector level, Eq. 3 can be interpreted as:

$$\mathbf{e}_u^{(l)} = f_{combine}^{(l)}(\mathbf{e}_u^{(l-1)}, f_{aggregate}^{(l)}(\{\mathbf{e}_i^{(l)} | i \in \mathcal{N}_u\})), \tag{4}$$

$$\mathbf{e}_i^{(l)} = f_{combine}^{(l)}(\mathbf{e}_i^{(l-1)}, f_{aggregate}^{(l)}(\{\mathbf{e}_u^{(l)} | u \in \mathcal{N}_i\})), \tag{5}$$

where $\mathcal{N}_u$ and $\mathcal{N}_i$ are the neighbor node set of user $u$ and item $i$, respectively. There are many works designing different $f_{combine}$ and $f_{aggregate}$ [5, 26, 35]. Usually, there will be readout function to generate the final representations for the recommendation task:

$$\mathbf{e} = f_{readout}(\{\mathbf{e}^{(l)} | l = 0, 1, \ldots, L\}). \tag{6}$$

For example, $f_{readout}$ can be concatenation [30], weighted sum [8] and retaining the last output [24].

**LightGCN Brief.** In this paper, we implement our CGI on the simple but effective GCN-based recommendation model LightGCN. It adopts weighted sum aggregators and abandon the use of feature transformation and nonlinear activation, of which the matrix form can be formulated as:

$$\mathbf{E}^{(l)} = (\mathbf{D}_{\mathcal{G}}^{-\frac{1}{2}} \mathbf{A}_{\mathcal{G}} \mathbf{D}_{\mathcal{G}}^{-\frac{1}{2}}) \mathbf{E}^{(l-1)}, l \in \mathbb{N}^+, \tag{7}$$

where $\mathbf{E}^{(l-1)} = [\mathbf{E}_u^{(l-1)}, \mathbf{E}_i^{(l-1)}]$ is the output of the previous LightGCN layer or the initial $\mathbf{E}^{(0)}$. At last, LightGCN implement the $f_{readout}$ by weighted sum, in which the weight of each layer is set as $\frac{1}{L+1}$ following the original work.

After obtaining the representations of users and items, the inner product $\hat{r}_{ui} = \mathbf{e}_u^T \mathbf{e}_i$ is used to predict preference score, which is commonly adopted in most recommender system: LightGCN employ the Bayesian Personalized Ranking (BPR) loss [17] to optimize the model parameters: $\mathcal{L}_{rec} = \sum_{(u,i,j) \in \mathcal{O}} -ln\sigma(\hat{r}_{ui} - \hat{r}_{uj})$, where $\mathcal{O} = \{(u,i,j) | (u,i) \in \mathcal{R}^+, (u,j) \in \mathcal{R}^-\}$ is the pairwise training data, in which $\mathcal{R}^+$ denotes the observed interactions, and $\mathcal{R}^-$ denotes the unobserved interactions. In this work, we also choose it as the objective function for the recommendation task.

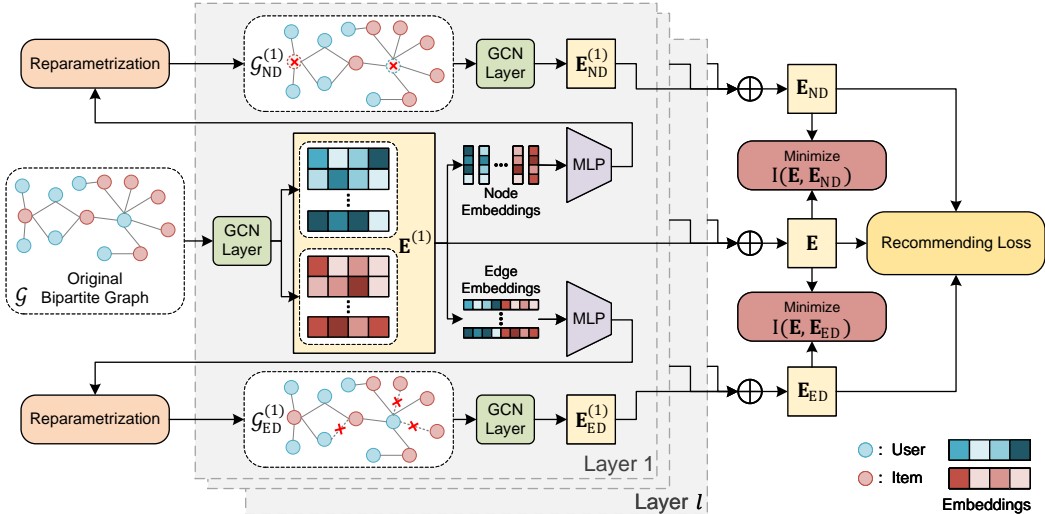

Figure 2: The overview the CGI framework. We integrate both the node-dropping and edge-dropping views together for a more comprehensive representation, though they can be applied separately.

## 4 Methodology

The framework of CGI is illustrated in Fig. 2 and we detail the inference in Appendix.

### 4.1 Learnable Multi-View Augmentation

Most of GCN-based recommendation like LightGCN [8] fully relies on the adjacency matrix $\mathbf{A}_{\mathcal{G}}$ to refine the representations of users and items in Eq. 7. However, $\mathbf{A}_{\mathcal{G}}$ may contain many biased and noisy information as discussed in Sec. 1, which continue to propagate misleading information as the LightGCN goes deeper. On the other hand, the vanilla randomly dropout in most contrastive learning for recommendation cannot create powerful views to alleviate popularity bias and interaction noises. We hence utilize parameterized networks to generate the layer-wise optimized augmentation views. Specifically, we assign different graph convolution layers with different learned subgraphs coupled with the downstream recommendation and thus obtain multi-view user and item representations. We elaborate on two types of learnable augmentations as follows.

**Node-Dropping View**  As illustrated in Sect. 1, popular users or items in the graph may skew the data distribution and thus hinder the GCN-based recommender. So we perform learnable node dropping at each layer to mask those the influential nodes and create the Node-Dropping view, which can be formulated as:

$$\mathcal{G}_{ND}^{(l)} = \{\{v_i \odot \rho_i^{(l)} \mid v_i \in \mathcal{V}\}, \mathcal{E}\}, \tag{8}$$

where $\rho_i^{(l)} \in \{0, 1\}$ is drawn from a Bernoulli distribution parameterized by $\omega_i^{(l)}$, i.e., $\rho_i^{(l)} \sim Bern(\omega_i^{(l)})$, which denotes whether to keep the node $v_i$.

Simply removing the selected node alongside all its connections will cause a dramatic change of the bipartite graph structure thus exerting influence on the information aggregation and making the training unstable. Thus instead of removing the selected node, we replace the selected node $v$ with its local subgraph's representation to obscure its original representation and retain its corresponding edges. For node $v$, we perform random walk on the bipartite graph $\mathcal{G}$ with its walk length setting as $k$, then we take the mean pooling of sampled nodes as $v$'s local subgraph's representation.

**Edge-Dropping View**  The goal of the Edge-Dropping view is to generate a subgraph filtering out noisy edges and intentionally decreasing the influence of popular nodes for GCN layers. Similarly to the Node-Dropping view, we create the Edge-Dropping view by learnable edge dropping:

$$\mathcal{G}_{ED}^{(l)} = \{\mathcal{V}, \{e_{ij} \odot \rho_{ij}^{(l)} \mid e_{ij} \in \mathcal{E}\}\}, \tag{9}$$

where $\rho_{ij}^{(l)} \in \{0, 1\}$ also follows $\rho_{ij}^{(l)} \sim Bern(\omega_{ij}^{(l)})$ and denotes whether the edge $e_{ij}$ is present.

Following [26], we adopt multi-layer perceptrons (MLPs) to the parameter $\omega_i^{(l)}$ and $\omega_{ij}^{(l)}$ that control the whether to mask node $v_i$ and edge $e_{ij}$, respectively, which can be formulated as:

$$\omega_i^{(l)} = MLP(\mathbf{e}_i^{(l)}); \quad \omega_{ij}^{(l)} = MLP([\mathbf{e}_i^{(l)}; \mathbf{e}_j^{(l)}]). \tag{10}$$

To efficiently optimize the multi-view structure learning in an end-to-end manner, we adopt the reparameterization trick [11] and relax the above binary entries $\rho$ from being drawn from Bernoulli distribution to a deterministic function of parameter $\omega$ and an independent random variable $\epsilon$, which can be formulate as:

$$\rho = \sigma((\log \epsilon - \log(1 - \epsilon) + \omega)/\tau), \tag{11}$$

where $\epsilon \sim Uniform(0, 1)$, $\tau \in \mathbb{R}^+$ indicates the temperature and $\sigma(\cdot)$ is the sigmoid function. With $\tau > 0$, the function is smoothed with a well-defined gradient $\frac{\partial \rho}{\partial \omega}$, enabling efficient optimization of the learnable establishment of Node-Dropping view and Edge-Dropping view during training. In inference, we drop the node or edge with a probability of less than 0.5.

Afterwards, we perform GCNs to obtain the representation of users and items on these views:

$$\mathbf{E}_{ND}^{(l)} = GCN(\mathbf{E}_{ND}^{(l-1)}, \mathcal{G}_{ND}^{(l)}), \quad \mathbf{E}_{ED}^{(l)} = GCN(\mathbf{E}_{ED}^{(l-1)}, \mathcal{G}_{ED}^{(l)}), \tag{12}$$

where the initial $\mathbf{E}_{ND}^{(0)} = \mathbf{E}_{ED}^{(0)} = \mathbf{E}^{(0)}$. After stacking $L$ LightGCN layers, we also adopt the weighted sum to construct their final representation $\mathbf{E}_{ND}$ and $\mathbf{E}_{ED}$, respectively. For simplicity, we omit the augmentation type $ND$ and $ED$ in the symbols below, and use $\tilde{\mathbf{E}}$ to denote the representations of these augmentation views.

## 4.2 Information Bottleneck Contrastive Learning

Although we couple the learnable augmentation process and the recommendation process together, we find relying solely on the recommendation objective can not well guide the dropout process to create optimal augmentation views. Thus we adopt the Information-Bottleneck principle to retain the **minimum sufficient information** in each view for the downstream recommendation. Specifically, different from conventional contrastive learning, we instead encourage the divergence between the representations of the augmentation view and the original graph while maximizing the information relevant to the recommendation task. By doing so, we can obtain comprehensive multi-view representation and efficiently drop noisy collaborative information for the recommendation. Accordingly, the objective in Eq. 1 is induced as:

$$\min_{(\mathbf{E}, \tilde{\mathbf{E}})} \tilde{\mathcal{L}}_{rec} + I(\mathbf{E}; \tilde{\mathbf{E}}), \tag{13}$$

where $\tilde{\mathcal{L}}_{rec}$ is the BPR loss of the representation from the augmentation view and $I(\mathbf{E}, \tilde{\mathbf{E}})$ represents the mutual information between representations from two corresponding views.

According to [25, 19], minimizing the InfoNCE loss [4] is equivalence to maximizing the lower bound of the corresponding mutual information. So we adopt negative InfoNCE to estimate the mutual information between the representations of the augmentation view and the original graph, which consists of mutual information from both the user side and item side. Formally, for the user side mutual information, we consider the representations of the same users in the augmentation view and the original graph as the positive pairs (i.e., $\{(\mathbf{e}_i, \tilde{\mathbf{e}}_i) \mid v_i \in \mathcal{U}\}$), while representations of two different users in the augmentation view and the original graph as the negative pairs (i.e., $\{(\mathbf{e}_i, \tilde{\mathbf{e}}_j) \mid v_i, v_j \in \mathcal{U}, i \neq j\}$):

$$I(\mathbf{E}_u; \tilde{\mathbf{E}}_u) = \sum_{v_i \in \mathcal{U}} \log \frac{exp(s(\mathbf{e}_i, \tilde{\mathbf{e}}_i)/\tau')}{\sum_{v_j \in \mathcal{U}} exp(s(\mathbf{e}_i, \tilde{\mathbf{e}}_j)/\tau')}, \tag{14}$$

where $s(\cdot)$ measures the similarity between two vectors, which is set as cosine similarity function; $\tau'$ is the hyper-parameter indicating the temperature similar to Eq. 11. Analogously, we can obtain the mutual information from item side $I(\mathbf{E}_i; \tilde{\mathbf{E}}_i)$ and the overall mutual information can be obtained by combining mutual information from two sides: $I(\mathbf{E}; \tilde{\mathbf{E}}) = I(\mathbf{E}_u; \tilde{\mathbf{E}}_u) + I(\mathbf{E}_i; \tilde{\mathbf{E}}_i)$.

## 4.3 Optimization

To obtain comprehensive multi-view representations, we utilize two parameterized networks to learn to create the Node-Dropping view and the Edge-Dropping view simultaneously. In order to integrally explore both views for better recommendation, we jointly optimize the recommendation tasks of these views and the self-supervised IB contrastive learning:

$$\mathcal{L} = \mathcal{L}_{rec} + \mathcal{L}_{rec}^{ND} + \mathcal{L}_{rec}^{ED} + \lambda(I(\mathbf{E}, \mathbf{E}_{ND}) + I(\mathbf{E}, \mathbf{E}_{ED})) + \beta\|\Theta\|_2^2, \tag{15}$$

where $\mathcal{L}_{rec}^{ND}$ and $\mathcal{L}_{rec}^{NB}$ are the recommendation objective of the Node-Dropping view and Edge-Dropping view respectively. The last term is an $L_2$ regularization. $\lambda$ and $\beta$ are the hyper-parameters controlling the effect strength of the IB contrastive learning task and $L_2$ regularization, respectively.

**Proposition 1.** Formally, we denote the learned augmentation view as $\tilde{\mathcal{G}}$, the noisy graph structure as $\mathcal{G}'$, and the downstream recommendation information as $Y_{Rec}$. Suppose $\mathcal{G}'$ is irrelevant to $Y_{Rec}$, the mutual information $I(\mathcal{G}'; \tilde{\mathcal{G}})$ is upper bounded by $I(\mathcal{G}; \tilde{\mathcal{G}}) - I(Y_{Rec}; \tilde{\mathcal{G}})$:

$$I(\mathcal{G}'; \tilde{\mathcal{G}}) \leq I(\mathcal{G}; \tilde{\mathcal{G}}) - I(Y_{Rec}; \tilde{\mathcal{G}}). \tag{16}$$

*Proof.* Following the Markov chain assumption in [1], we suppose $\mathcal{G}$ is defined by $Y$ and $\mathcal{G}'$. And we can define the following Markov chain $(Y_{Rec}, \mathcal{G}') \to \mathcal{G} \to \tilde{\mathcal{G}}$. According to the Data Processing Inequality, we have:

$$\begin{aligned} I(\mathcal{G}; \tilde{\mathcal{G}}) \geq I((Y_{Rec}, \mathcal{G}'); \tilde{\mathcal{G}}) &= I(\mathcal{G}'; \tilde{\mathcal{G}}) + I(Y_{Rec}; \tilde{\mathcal{G}}|\mathcal{G}') \\ &= I(\mathcal{G}'; \tilde{\mathcal{G}}) + H(Y_{Rec}|\mathcal{G}') - H(Y_{Rec}|\mathcal{G}'; \tilde{\mathcal{G}}). \end{aligned} \tag{17}$$

Since $\mathcal{G}'$ and $Y_{Rec}$ are independent, we have $H(Y_{Rec}|\mathcal{G}') = H(Y_{Rec})$. Also, it's straightforward that $H(Y_{Rec}|\mathcal{G}'; \tilde{\mathcal{G}}) \leq H(Y_{Rec}|\tilde{\mathcal{G}})$. Thus we can simplify Eq. 17 as follow:

$$I(\mathcal{G}; \tilde{\mathcal{G}}) \geq I(\mathcal{G}'; \tilde{\mathcal{G}}) + H(Y_{Rec}) - H(Y_{Rec}|\tilde{\mathcal{G}}) = I(\mathcal{G}'; \tilde{\mathcal{G}}) + I(Y_{Rec}; \tilde{\mathcal{G}}). \tag{18}$$

Thus we obtain that $I(\mathcal{G}'; \tilde{\mathcal{G}}) \leq I(\mathcal{G}; \tilde{\mathcal{G}}) - I(Y_{Rec}; \tilde{\mathcal{G}})$, where $I(Y_{Rec}; \tilde{\mathcal{G}})$ is inverse proportional to the $\tilde{\mathcal{L}}_{rec}$ in Eq. 13. Eq. 16 proves that optimizing the IB contrastive objective in Eq. 13 is equivalent to minimizing the mutual information between the learned augmentation view and noisy structure. Specifically, it provides theoretical guarantees that the IB contrastive learning leads to the noise-invariance property by compressing the information in both the augmentation views. Meanwhile, the IB contrastive objective also restricts the augmentation view to be predictive for the recommendation task, which can intentionally reduce the influence of popular nodes while preserving information of the isolated nodes, and thus help to mitigate the popularity bias.

## 5 Experiments

### 5.1 Experimental Setup

**Dataset Description**   Three public available datasets are employed in our experiments, i.e., *Yelp2018*, *MovieLens-1M* and *Douban*. The detailed description can be found in the Appendix. For each dataset, we randomly select $80\%$ of the historical interactions of each user as the training set, $10\%$ of those as the validation set, and the remaining $10\%$ as the test set.

**Evaluation metrics**   To evaluate the performance of all methods, we adopt a ranking-based metric namely Normalized Discounted Cumulative Gain@$k$ (NDCG@$k$) and a relevancy-based metric Hit Ratio@$k$ (RECALL@$k$). The formulations of the two metrics are in the Appendix. As suggested by Krichene and Rendle [13], we perform item ranking on all the candidate items instead of the sampled item sets to calculate above metrics, which guarantees that the evaluation process is unbiased.

**Compared Methods**   We compare our CGI with three classes of baseline methods: (1) MF-based methods, i.e., **BPRMF** [17] and **NCF** [7], (2) GNNs-based methods, i.e., **NGCF** [30] and **LightGCN** [8], and (3) CL-based methods, i.e., **DNN+SSL** [36] and **SGL** [31]. We give a detailed introduction to these baselines in the Appendix. Note that DNN+SSL applies augmentation on items' feature which is not applicable in our case. So following [31], we apply the augmentations on ID embeddings of items instead.

Table 1: Comparison among models. Boldface denotes the highest score and underline indicates the best result of the baselines.

| Model | Yelp2018 | | | | MovieLens-1M | | | |
|---|---|---|---|---|---|---|---|---|
| | NDCG@10 | RECALL@10 | NDCG@20 | RECALL@20 | NDCG@10 | RECALL@10 | NDCG@20 | RECALL@20 |
| BPRMF | 0.0138 | 0.0209 | 0.0191 | 0.0373 | 0.1225 | 0.1376 | 0.1407 | 0.1882 |
| NCF | 0.0224 | 0.0356 | 0.0289 | 0.0566 | 0.1430 | 0.1546 | 0.1576 | 0.2027 |
| NGCF | 0.0242 | 0.0384 | 0.0319 | 0.0629 | 0.1462 | 0.1651 | 0.1667 | 0.2285 |
| LightGCN | 0.0344 | 0.0530 | 0.0445 | 0.0850 | 0.1696 | 0.1865 | 0.1863 | 0.2420 |
| DNN+SSL | 0.0217 | 0.0344 | 0.0286 | 0.0564 | 0.1096 | 0.1238 | 0.1250 | 0.1714 |
| SGL | 0.0367 | 0.0552 | 0.0473 | 0.0891 | 0.1800 | 0.1965 | 0.1972 | 0.2520 |
| CGI | **0.0392** | **0.0584** | **0.0501** | **0.0932** | **0.1979** | **0.2180** | **0.2152** | **0.2772** |
| *Improv.* | +6.82% | +5.90% | +5.93% | +4.58% | +9.95% | +10.91% | +9.13% | +9.97% |
| *p-value* | 1.29e-3 | 3.53e-3 | 7.00e-4 | 3.59e-4 | 8.89e-4 | 4.22e-4 | 4.83e-4 | 5.07e-5 |

| Model | Douban | | | |
|---|---|---|---|---|
| | NDCG@10 | RECALL@10 | NDCG@20 | RECALL@20 |
| BPRMF | 0.0496 | 0.0526 | 0.0516 | 0.0613 |
| NCF | 0.0694 | 0.0706 | 0.0659 | 0.0734 |
| NGCF | 0.0794 | 0.0823 | 0.0784 | 0.0897 |
| LightGCN | 0.0862 | 0.0876 | 0.0845 | 0.0940 |
| DNN+SSL | 0.0712 | 0.0738 | 0.0703 | 0.0804 |
| SGL | 0.0912 | 0.0906 | 0.0910 | 0.1012 |
| CGI | **0.0991** | **0.1007** | **0.0979** | **0.1119** |
| *Improv.* | +8.69% | +11.18% | +7.55% | +10.55% |
| *p-value* | 1.99e-3 | 4.40e-3 | 1.52e-4 | 1.60e-4 |

**Hyper-parameter**  We initialize the latent vectors of both users and items with small random values for all models. The parameters for baseline methods are initialized as in the original papers, and are then carefully tuned to achieve optimal performances. For a fair comparison, the dimensions of both the user and item embeddings are all fixed to 64. We use Adam with $\beta_1 = 0.9$, $\beta_2 = 0.999$, $\epsilon = 1e^{-8}$ to optimize all these methods. The batch size is set to 2048. The learning rate is set as 0.005 and decayed at the rate of 0.9 every five epochs. We set $\lambda = 0.02$ and $\beta = 0.01$ for the coefficients in Eq. 15. More details about hyper-parameter settings of baselines can be found in the Appendix.

## 5.2  Performance Comparisons

We summarize the performance of different algorithms in terms of NDCG@$k$ and RECALL@$k$ ($k = 10, 20$) over three datasets in Table 1. The experimental results demonstrate that CGI outperforms other methods on all evaluation metrics. We conduct the significant test and p-values $< 0.05$ indicates that the improvements of our CGI are statistically significant.

Besides, we observe that the GNNs-based methods perform better than the MF-based models. These results verify that exploiting higher-order connectivity in the user-item bipartite graph is essential to improve the recommendation performance. This may also be the reason why the performance of DNN+SSL is inferior to those of SGL and our CGI when all applying contrastive learning. We can see that the CL-based graph learning methods, including our CGI, consistently outperform the GNNs-based models, which verifies the effectiveness of contrastive learning for representation learning. Besides, our CGI outperforms SGL by a large margin. The results demonstrate that compared with randomly dropping in SGL, the learnable graph augmentations optimized by information bottleneck can create optimal augmentation views and capture more comprehensive collaborative signals.

## 5.3  Ablation Studies

**Effectiveness of Learnable Augmentation**  To understand the respective effects of both the node-dropping and edge-dropping in learnable augmentation, we conduct ablation studies on Yelp2018 and Movielens-1M. As shown in Table 2, we report NDCG@10 and RECALL@10 of CGI and SGL in different versions. Specifically, $CGI_{ND}$ and $CGI_{ED}$ denote CGI with only node-dropping view and edge-dropping view being adopted, respectively. $SGL_{ND}$ and $SGL_{ED}$ denotes the augmentation view in SGL is created by random node dropout and edge dropout, respectively.

We find that: (1) Our CGI achieves obvious improvements compared with SGL in different types of augmentation, which again verifies the effectiveness of the learnable graph augmentation optimized by information bottleneck. (2) CGI performs better in both CGI-ND and CGI-ED. We ascribe these to the ability of multi-view learning, which enables the final representation to capture collaborative information of different semantics and thus enhances the robustness and expressiveness of the model.

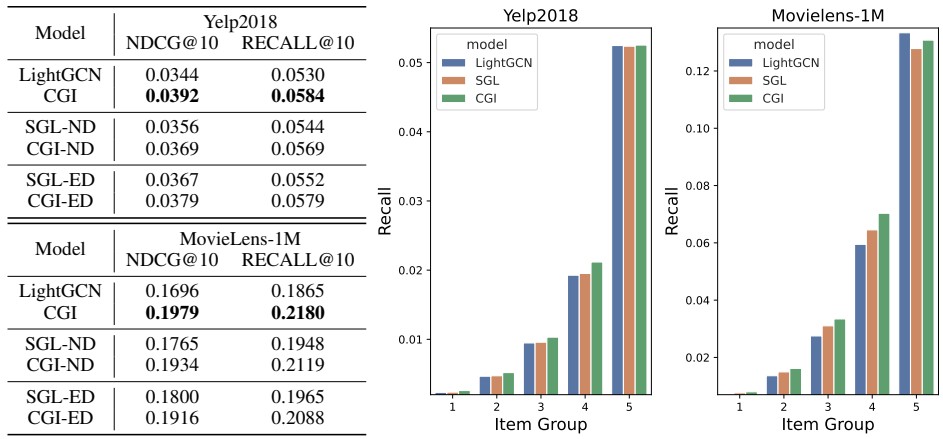

| Model | Yelp2018 | |
| --- | --- | --- |
| | NDCG@10 | RECALL@10 |
| LightGCN | 0.0344 | 0.0530 |
| CGI | **0.0392** | **0.0584** |
| SGL-ND | 0.0356 | 0.0544 |
| CGI-ND | 0.0369 | 0.0569 |
| SGL-ED | 0.0367 | 0.0552 |
| CGI-ED | 0.0379 | 0.0579 |

| Model | MovieLens-1M | |
| --- | --- | --- |
| | NDCG@10 | RECALL@10 |
| LightGCN | 0.1696 | 0.1865 |
| CGI | **0.1979** | **0.2180** |
| SGL-ND | 0.1765 | 0.1948 |
| CGI-ND | 0.1934 | 0.2119 |
| SGL-ED | 0.1800 | 0.1965 |
| CGI-ED | 0.1916 | 0.2088 |

Table 2: Comparison among models.

Figure 3: Performance of different item groups

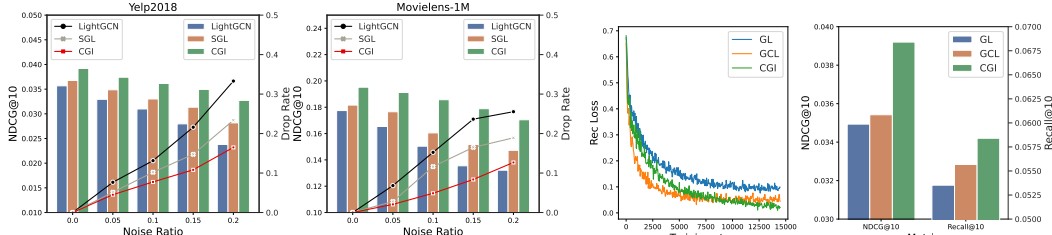

Figure 4: Performance comparison over different noise ratio. The bar represents the NDCG@10 and the line represent the performance degradation ratio.

Figure 5: Effect of Information Bottleneck on Yelp2018

(3) The performance of CGI-ED is better than that of CGI-ND in the sparse dataset Yelp2018, while worse in the dense dataset Movielens-1M. We can speculate that the interaction noises are more significant in the sparse dataset with less useful information, in which CGI-ND is not so flexible. Because it will remove all influence (i.e., edges) of popular nodes, which is hard to be restored with scarce interactions. But in the dense dataset, popularity bias becomes more significant, which makes CGI-ND more efficient by blocking the influence from popular users or items.

**Accuracy against Popularity Bias**   To verify whether CGI is capable of mitigating popularity bias, We split the item set $\mathcal{I}$ into 5 groups (1-5) evenly based on their popularity. The larger the GroupID is, the larger degrees the items have. Following [31], we decompose the RECALL@10 metric of the whole dataset into the contributions of the above ten groups of items:

$$\text{RECALL}^{(g)} = \frac{\sum_{i=1}^{k} rel_i^{(g)}}{|\mathcal{I}_{test}^u|} , \tag{19}$$

where $rel_i^{(g)} = 1$ denotes the item at the rank $i$ is in the test set and $g$-th item group at the same time. As such, RECALL$^{(g)}$ measures the performance over the $g$-th item group. From Fig. 3, we can see that recommender systems tend to recommend popular items, while leaving unpopular items less likely to be discovered, which further exacerbates the long-tail distribution. Also, our CGI can significantly improve the recommendation accuracy on long-tail items. Although both GCL methods CGI and SGL, show no superiority on the top 20% items, from the overall improvements in Table 1, we can see they can better capture the long-tail items' information in user preference representations.

**Robustness to Interaction Noises**   To verify CGI's robustness to interaction noises, we generate different proportions of negative interactions (i.e., 5%, 10%, 15%, and 20%) to contaminate the training set, and report the performance on the unchanged test set. Fig. 4 shows the NDCG@10 on Yelp2018 and Movielens-1M and the performance degradation ratio of the corresponding contaminated training set. It's obvious that the more noise we add, the worse performance all the models yield, since all the models utilize LightGCN as the basic backbone, which fully relies on the adjacency matrix $\mathbf{A}_{\mathcal{G}}$ to refine the representations of users and items in Eq. 7. However, the performance degradation of

Table 3: Performance with Other GNN variants.

| Model | Yelp2018 | | Movielens-1M | | Douban | |
|---|---|---|---|---|---|---|
| | NDCG@10 | Recall@10 | NDCG@10 | Recall@10 | NDCG@10 | Recall@10 |
| GC-MC | 0.0214 | 0.0278 | 0.1350 | 0.1491 | 0.0671 | 0.0739 |
| SGL+GC-MC | 0.0218(+1.9%) | 0.0281(+1.2%) | 0.1412(+4.6%) | 0.1577(+5.8%) | 0.0687(+2.3%) | 0.0762(+3.1%) |
| CGI+GC-MC | 0.0218(+2.1%) | 0.0282(+1.7%) | 0.1422(+5.3%) | 0.1585(+6.3%) | 0.0687(+2.3%) | 0.0765(+3.5%) |
| NGCF | 0.0242 | 0.0384 | 0.1462 | 0.1651 | 0.0794 | 0.0823 |
| SGL+NGCF | 0.0260(+7.4%) | 0.0418(+8.9%) | 0.1609(+10.1%) | 0.1871(+13.3%) | 0.0833(+4.9%) | 0.0857(+4.1%) |
| CGI+NGCF | 0.0272(+12.5%) | 0.0431(12.1%) | 0.1660(%13.6%) | 0.1937(+17.3%) | 0.0840(+5.7%) | 0.0875(+6.3%) |

our CGI is smaller than other models in both datasets. What's more, the gaps between CGI and other models grow larger as the noise increase. This suggests that our CGI framework can mitigate the noise in interaction data more efficiently, and our learnable augmentation optimized by the IB contrastive learning exhibits good robustness in the presence of a high proportion of noise, which is consistent with our proof in Sect. 4.3. We can observe that CGI is more robust on Movielens-1M. This makes sense since Movielens-1M is much denser than Yelp2018 according to the statistics in the Appendix and thus the bipartite graph of Yelp2018 will be more sensitive to the added noise.

**Effectiveness of Information Bottleneck** To investigate the effect of information bottleneck, we consider the following variants of CGI with different contrastive learning strategies, our complete methods (CGI), our method without introducing contrastive learning (GL), and our method that maximizes the correspondence among different views (i.e., $\min \tilde{\mathcal{L}}_{rec} - I(\mathbf{E}; \tilde{\mathbf{E}})$) (GCL). Fig. 5 shows the recommending training loss *w.r.t.* the number of training steps and the evaluation results on Yelp, from which we observe that the multi-view graph learning frameworks driven by contrastive learning are easier to converge. Specifically, when maximizing the mutual information among views, the GCL framework drops more quickly at the very beginning and turns to a steadily decreasing state afterward. However, with IB contrastive learning, the recommending loss of our CGI appears to have a declining trend after an initial sharp drop, instead of getting an early-stop, which is more likely to converge to a better local optimum. This is probably why **CGI** has better performance than both **GL** and **GCL**, as illustrated by the right part of Fig. 5. Also, we find that the multi-view graph learning can benefit more from the IB contrastive learning than the conventional one, since it can encourage to drop the noisy information irreverent for the recommendation as illustrated in Sect. 4.3.

**Performance with Other GNNs** To verify the generalization of our method on other GNNs, we tried CGI and the baseline SGL on two other popular GNN-based recommenders GC-MC [24] and NGCF [30]. The experimental results are shown in Table 3. Both graph contrastive learning methods have shown improvements to the backbones. On NGCF, CGI shows consistent superiority compared to SGL. On GC-MC, CGI does not have significant improvement compared to SGL, probably due to the fact that GC-MC only utilizes one layer of GCN, which means it can only adopt 1-hop neighbors for learning, thus making the learnable augmentation challenging to fetch enough information.

## 6 Conclusions

In this paper, we propose novel Contrastive Graph Structure Learning via Information Bottleneck (CGI) to learn better augmentation from different aspects for the multi-view representation learning of recommendation. In particular, we propose a fully differentiable learner to drop nodes and edges to construct different types of augmentation views coupled with the recommendation. We innovatively integrate information bottleneck into the multi-view contrastive learning process for recommendation and prove its efficiency. The extensive experiments conducted on three public datasets verify the effectiveness of CGI.

## Acknowledgments and Disclosure of Funding

This work was supported by Alibaba Group through Alibaba Research Intern Program.

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
