# Appendix: Contrastive Graph Structure Learning via Information Bottleneck for Recommendation

**Chunyu Wei**[1][*]**, Jian Liang**[1][*]**, Di Liu**[1]**, Fei Wang**[2]
[1]Alibaba Group, China
[2]Department of Population Health Sciences, Weill Cornell Medicine, USA
weicy15@icloud.com
{xuelang.lj, wendi.ld}@alibaba-inc.com
few2001@med.cornell.edu

## A   Inference

At inference time, in order to integrate the rich information of different semantics from different views, we concatenate the representations from the original view and the augmentation views, which can be formulated as follows:

$$\mathbf{E}_{infer} = \mathbf{E} \parallel \mathbf{E}_{ND} \parallel \mathbf{E}_{ED}, \tag{1}$$

where $\parallel$ denotes concatenation. Note that other functions like mean-pooling, max-pooling can be adopted, but we found that concatenation performs better when utilizing LightGCN as the backbone.

## B   Degree Distribution of Dropping Structures

GCNs rely on message-passing mechanism and aggregate the information from neighbors to learn representative embeddings of users and items. We believe that random dropout will indiscriminately drop nodes or edges regardless of the corresponding node degrees, while by message passing mechanism, GCNs are easier to reconstruct the missing information of popular users or items, but much harder to reconstruct those isolated nodes with few connections, thus may overemphasize those high-degree nodes. So we proposed learnable graph augmentation to intentionally reduce the influence of popular nodes while preserving information of the isolated nodes. To verify its effeteness, we split the node set $\{v_i \mid v_i \in \mathcal{V}\}$ and the edge set $\{e_{ij} \mid e_{ij} \in \mathcal{E}\}$ into five groups evenly based on the node degrees $Degree_{v_i}$ and degree sum of corresponding nodes $Degree_{v_i} + Degree_{v_j}$. A large GroupID denotes the node or edge lies in a dense structure of the graph. We report the proportions of different degree groups of the dropping nodes or edges in the last GCN layer. If we adopt the random dropout, each group will maintain 20%, since we split the node/edge set evenly. However from Figure 1, we find that, as the degree becomes larger, more node/edge will be dropped in our model. This proves that our model tends to drop those dense structures when we encourage the augmentation to be different from the original graph.

## C   Experiment on Popularity Bias

We divide the items into the top 20% head items and the last 80% tail items based on their popularity, as they follow the long tail distribution. We report statistics about the head items and tail items in the top-10 recommendation lists of all users by different baselines as Table 1. From these statistics in Table 1, we can observe that tail items have more chance to be recommended by CGI than in other baselines, which verifies the ability of CGI to alleviate the popularity bias. In addition, we have conducted quantitative and qualitative experiments on how rare items are ranked by the different

---

[*]Equal contributions from both authors. This work is done when Chunyu Wei works as an intern at Alibaba.

36th Conference on Neural Information Processing Systems (NeurIPS 2022).

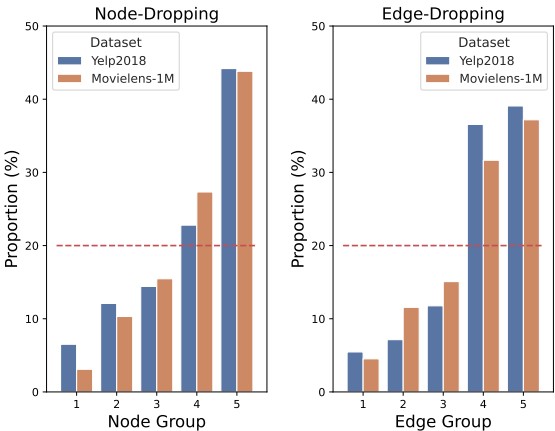

Figure 1: Degree Distribution of Dropping Structures

Table 1: Statistics of Recommendation Results

|  | Yelp2018 | | Movielens-1M | | Douban | |
|---|---|---|---|---|---|---|
|  | Head | Tail | Head | Tail | Head | Tail |
| LightGCN | 97.19% | 2.81% | 96.11% | 3.89% | 97.41% | 2.59% |
| SGL | 93.41% | 6.59% | 94.89% | 5.11% | 94.97% | 5.03% |
| CGI | 92.72% | 7.28% | 93.79% | 6.21% | 93.83% | 6.17% |

models for a given set of users following [4]. In the Movielens-1M, we randomly select two users and rank all movies for the users with different models. We divide the ranking results into ten groups and show the numbers of rare movies (with less than 20 watches) in each ranking group. The results are presented in Fig. 2, where each subfigure represents one user. From the histograms, we can observe

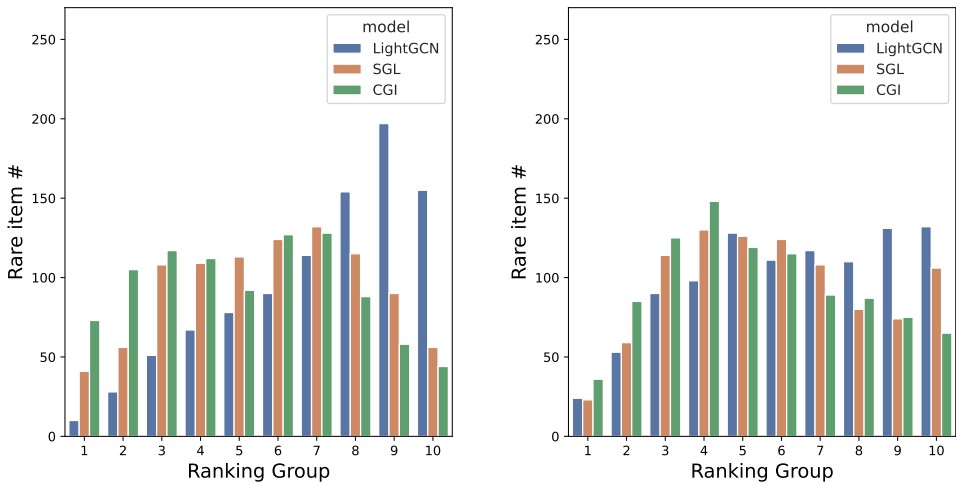

Figure 2: Numbers of Rare Movies in Each Ranking Group

that CGI is capable of pushing rare items to the top ranks of the recommendation.

## D   Limitation and Potential Negative Societal Impacts

In this work, we reveals the limitations of graph-based recommendation and attempts to exploit learnable graph augmentation and information bottleneck to enhance the contrastive learning for

recommendation. Despite the significant improvement, a limitation of the work is in the type of feedback considered. We only consider the "purchasing" action of users towards items to construct the bipartite graph. However, a platform could have access to other types of signals ("like", "add to basket", "add to wishlist", "remove from wishlist", etc.) and we plan to explore new perspectives, such as integrating multiple types of signals, to create more powerful data augmentations for a heterogeneous graph.

This method exerts a positive influence on the society and the community, opens up new research possibilities, promotes fair exposure of e-commerce, and boost online users' satisfaction level. However, this work suffers from some negative consequences, which is worthy of further research and exploration. Specifically, more jobs of new product promotion and customer survey may be cancelled. Besides, we should be cautious of the result of the failure of the system, which could render people to buy some products of poor quality.

## E  Time Complexity

Suppose the number of nodes and edges in the user-item bipartite graph $\mathcal{G}$ are $|\mathcal{V}|$ and $|\mathcal{E}|$ respectively. Let $L$ denote the number of GCN layers, $B$ denote the mini-batch size, $s$ denote the number of epochs and $d$ is the embedding size. The complexity of CGI during model training consists of four parts. (1) According to [2], the complexity of the graph convolution on the full bipartite graph is $O(2|\mathcal{E}|dLs\frac{|\mathcal{E}|}{B})$. When we integrate both the node-dropping view and edge-dropping view, the total complexity of the graph convolution part is $O(6|\mathcal{E}|dLs\frac{|\mathcal{E}|}{B})$. (2) As for the learnable multi-view augmentation, since we adopt a two-layer MLP to learn the Bernoulli parameter, the complexity for learning the node-dropping view is $O(|\mathcal{V}|d^2Ls\frac{|\mathcal{E}|}{B})$, while complexity for learning the edge-dropping view is $O(2|\mathcal{E}|d^2Ls\frac{|\mathcal{E}|}{B})$. (3) The BPR recommending loss for the original graph and two subgraphs is $O(6|\mathcal{E}|ds)$. (4) As for the IB contrastive learning, we treat all the other users in $\mathcal{U}$ as negative samples in Eq. 2 when computing the mutual information of the user side.

$$I(\mathbf{E}_u; \tilde{\mathbf{E}}_u) = \sum_{v_i \in \mathcal{U}} \log \frac{exp(s(\mathbf{e}_i, \tilde{\mathbf{e}}_i)/\tau')}{\sum_{v_j \in \mathcal{U}} exp(s(\mathbf{e}_i, \tilde{\mathbf{e}}_j)/\tau')}, \tag{2}$$

In practice, we only treat the users in the same batch as negative samples to reduce the complexity. So the complexity of the user side per epoch would be $O(B(d + Bd)\frac{|\mathcal{E}|}{B}) = O(d(B+1)|\mathcal{E}|)$. Similarly, it would also be $O(d(B+1)|\mathcal{E}|)$ for the item side. So the total complexity for IB contrastive learning would be $O(2d(B+1)|\mathcal{E}|s)$.

From the analyses above, we can find that the heaviest computation, in theory, is from the learning of edge-dropping view, i.e., $O(2|\mathcal{E}|d^2Ls\frac{|\mathcal{E}|}{B})$, since usually $|\mathcal{V}| < |\mathcal{E}|$. So the analytical complexity of CGI actually scales the complexity of LightGCN with embedding size $d$, which is actually in the same magnitude with conventional GCNs with feature transformation, e.g., NGCF [5]. It's worth noting that for inference, we don't need to perform the IB contrastive learning in part (4), which further reduces the overall time complexity.

## F  Details of the Datasets

Here we describe the details of the datasets for experiments. The statistical details of these datasets are presented in Table 2.

**Yelp2018**  [2] This is an online location-based review system, on which users can express their experience (i.e., local businesses) through the form of reviews and ratings. We construct our dataset by using each review as evidence that a user's consumption. We filtered out users who post less than 10 reviews and businesses that receive less than 10 reviews.

**Movielens-1M**  [3] This is s a widely used dataset for recommendations, which contains one million user-movie ratings. In our case, we transform explicit ratings into implicit feedback, where each entry is viewed as 1 only when the rating is greater than 3.

---

[2]Data set available from https://www.yelp.com/dataset
[3]Data set available from https://grouplens.org/datasets/movielens/1m/

Table 2: Descriptive statistics of the datasets.

| Dataset | #Users | #Items | #Interactions | Density |
|---------|--------|--------|---------------|---------|
| Yelp2018 | 30,934 | 10,048 | 412,759 | 0.00133 |
| MovieLens-1M | 6,040 | 3,900 | 1,000,209 | 0.00425 |
| Douban | 2,848 | 15,171 | 65,789 | 0.00152 |

**Douban [8]**   This is a popular site on which users can review movies, music, and books they consume. Similar to Movielens-1M, this dataset is based on explicit ratings. So we also leave out ratings less than 4 and assign 1 to the rest. We retain users and items with at least five interactions.

## G   Formulations of Metrics

The NDCG@$k$ metric accounts for the position of the hits by assigning higher scores to hits at top ranks. The RECALL@$k$ metric measures the percentage of relevant items selected out of all the relevant items for the user. Both of the adopted metrics can be formulated as follows:

$$\text{NDCG@}k = \frac{1}{R_N} \sum_{i=1}^{N} \frac{2^{rel_i - 1}}{\log_2(1+i)} \,, \tag{3}$$

$$\text{RECALL@}k = \frac{\sum_{i=1}^{k} rel_i}{|\mathcal{I}_{test}^u|} \,, \tag{4}$$

where $k$ is the size of the recommendation list, $rel_i = 0$ or 1 denotes whether the item at the rank $i$ is in the test set or not, and the $R_N$ term indicates the maximum possible cumulative component through ideal ranking. $|\mathcal{I}_{test}^u|$ is the number of relevant items in the testing set for user $u$.

## H   Introduction to Baselines

Here we give a brief introduction to the baseline methods.

- **NCF** [1]. This is a deep learning based framework that combines matrix factorization (MF) with a multilayer perceptron model (MLP) for item ranking.
- **NGCF** [5]. This is a graph-based model, which first encodes the collaborative signal into the user-item interaction graph structure and adopts multiple graph convolution to explore high-order connectivity.
- **LightGCN** [2]. This is the state-of-the-art GCN-based general recommendation model that leverages the user-item proximity to learn node representations and generate recommendations.
- **DNN+SSL** [7] This is a self-supervise contrastive recommendation method, which utilizes DNNs as the encoder of items and adopts feature masking and feature dropout on the pre-existing features of items. Since we include no item feature in our case, we apply the augmentations on ID embeddings of items instead.
- **SGL** [6]. This is the state-of-the-art graph contrastive learning based recommendation method, which proposes randomly node dropout, edge dropout, and random walk for augmentation on the bipartite graph. Following the original paper, we adopt edge dropout on the LightGCN to obtain its best performance.

## I   Experiment on Douban

We repeat every experiment in the paper on **Douban**. For effects of both the node-dropping and edge-dropping in learnable augmentation, we add experiments on **Douban** and show results in Table 3. The experimental result for mitigating popularity bias on **Douban** is shown in Fig. 3. For robustness to interaction noises, the result on **Douban** is illustrated in Fig. 4. And we also investigate different contrastive learning strategies on **Douban** and present the result in Fig. 5. These supplemented experiments show great consistency with existing experimental results in the paper, which further supports the conclusions we made.

| Model | Douban | |
| | NDCG@10 | RECALL@10 |
| --- | --- | --- |
| LightGCN | 0.0862 | 0.0876 |
| CGI | **0.0991** | **0.1007** |
| SGL-ND | 0.0835 | 0.0823 |
| CGI-ND | 0.0903 | 0.0895 |
| SGL-ED | 0.0912 | 0.0906 |
| CGI-ED | 0.0965 | 0.0982 |

Table 3: Comparison among models on **Douban**

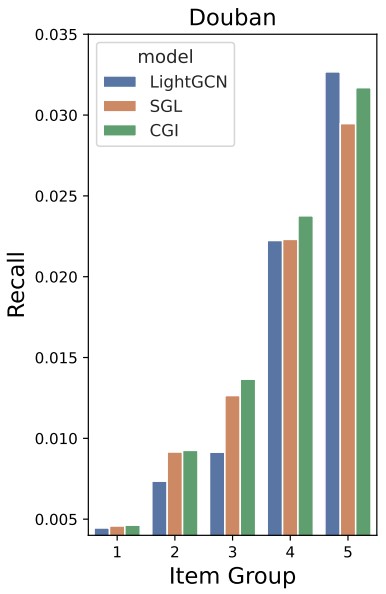

Figure 3: Performance of different item groups on **Douban**

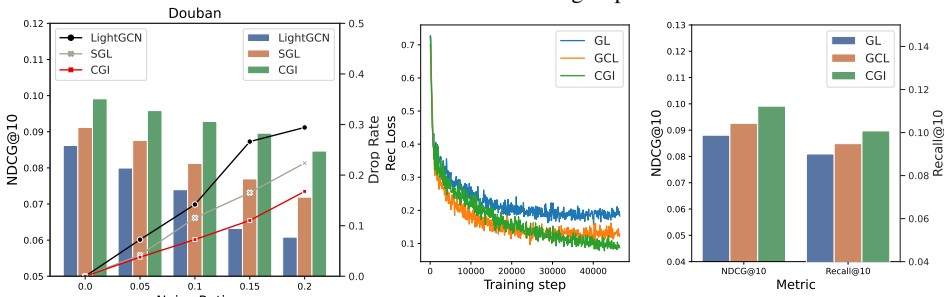

Figure 4: Performance over different noise ratio on **Douban**

Figure 5: Effect of Information Bottleneck on **Douban**

## J  Hardware and implementations in experiments

All experiments are conducted on a Linux machine with an Nvidia GeForce RTX 3090. CUDA version is 11.1 and Driver Version is 455.45.01. CGI is implemented with Pytorch. The learning rate for all models are tuned amongst $[0.005, 0.01, 0.02, 0.05]$. To prevent overfitting, we add $L_2$ norm with coefficient tuned from $[0.001, 0.005, 0.01, 0.02, 0.1]$. We select the best models by early stopping when the HR@20 on the validation set does not increase for three consecutive epochs.

## K  Parameter Studies

As verified in Sect. 4.1, the temperature $\tau$ in Equ. 5 plays an important role in the learnable augmentation.

$$\rho = \sigma((\log \epsilon - \log (1 - \epsilon) + \omega)/\tau), \qquad (5)$$

Specifically, for learning, there is a tradeoff between small temperatures, where the obtained entries $\rho$ are close to one-hot but the variance of the gradients is large, and large temperatures, where the reverse applies. So following [3], we start at 10 and anneal to 0.1 in the training process. Then we investigate the influence of parameter $\tau'$ in Eq. 2 and random walk length $k$ for constructing the node subgraph. We vary $\tau'$ from 0.1 to 1 and $k$ from 10 to 100, respectively, while keeping other parameters fixed. The results of HR@10 on *Yelp2018* and *Movielens-1M* are presented in Fig. 6. We can see that CGI achieves the best performance when $\tau' = 0.2$ in our scenario and either increasing or decreasing $\tau'$ will hurt the model performance. So we suggest to tune $\tau'$ in the range of $[0.1, 1.0]$. Also, we observe from Fig. 6 that, with the increase of $k$, the performance is boosted at first since a

larger group of neighbors can bring more accurate subgraph representation. However, it drops when $k$ reaches 20 in *Yelp2018* and 80 in *Movielens-1M*, because it may bring too much noise irrelevant to the local graph structure. Intuitively, the denser dataset requires a larger random walk length $k$, which is consistent with our observations.

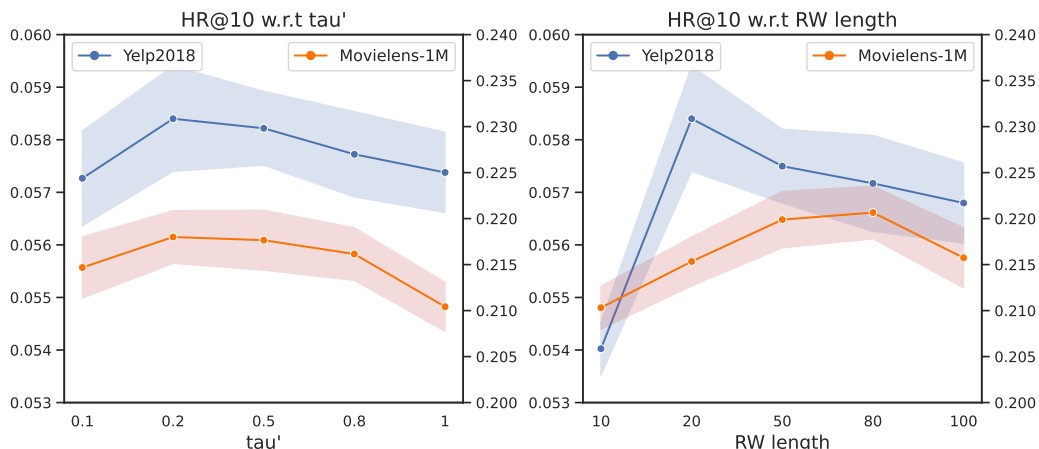

Figure 6: Parameter Studies of CGI