# OpenReview forum: "Contrastive Graph Structure Learning via Information Bottleneck for Recommendation"
_NeurIPS.cc/2022/Conference — NeurIPS 2022 Accept_

### Official Review · Reviewer_wqw8 · 2022-06-17

**Rating:** 8
**Confidence:** 3
**Soundness:** 4 excellent
**Presentation:** 4 excellent
**Contribution:** 3 good

**Summary:**

This paper presents a new deep learning architecture for graphs, named CGI. CGI aims at build a vector space for users and items that can be used for recommander systems. The main novelty lies in the way two views are built (one for the nodes, one for the edges), by using data augmentation with perturbation (for dealing with high-degree nodes) and Information Bottleneck framework to make the representation space more robust to noise. CGI is favorably compared to several recent models on three different datasets with a proper experimental framework (good metrics, ablation studies, etc.). I don't see any import flaw in the methodology. All in all, it seems to be as a good, rational contribution to the field.

**Questions:**

Is it possible to design a more accurate baseline by mitigating the weight of node centrality (like a normalization) before using solutions from the literature?

Section 4.3 is not perfectly clear to me. I don't understand to what extent Proposition 1 is a real new contribution (that can be used for other research, and in what context), or if it's a rather straightforward derivation.

**Ethics Review Area:**

["I don’t know"]

**Limitations:**

I don't see any important limitation. I've not given the maximal score because it's really interesting but not fundamentally novel (like, "groundbreaking").

However, I'm bothered by the fact the authors don't point out any possible limitation... and I cannot believe that. From my opinion, a paper like this should tell when the model isn't fit to the problem and what may be the possible extensions. It's maybe a problem of the (possible) computation cost of optimization, or when the two views (node vs. edge) disagree?

**Strengths And Weaknesses:**

Strengths:
- new interesting model with good motivations
- good results when compared to recent sota

Weaknesses:
I don't see any important weakness.

---

> ### Author Response · Authors · 2022-08-02
> **Response to Reviewer wqw8**
>
> Thanks for your detailed reading and suggestions.
> We provide detailed answers to your questions below.
>
>
> >**Q1**:
> Is it possible to design a more accurate baseline by mitigating the weight of node centrality (like a normalization) before using solutions from the literature?
>
>
> **A1**:
> Thanks for your detailed reading. In fact, the backbone model LightGCN of our CGI (and SGL) adopts graph Laplacian to normalize the graph. It calculates a graph Laplacian matrix $\mathit{L} = \mathbf{D}_{\mathcal{G}}^{-\frac{1}{2}} \mathbf{A}_\mathcal{G} \mathbf{D}_\mathcal{G}^{-\frac{1}{2}}$ in Eq. 7. $\mathbf{D}_\mathcal{G}$ is the diagonal degree matrix, where the $i$-th diagonal element $\mathbf{D}_\mathcal{G}[i,i] = degree_i$ represents the degree of $i$.
>
> As such, the nonzero off-diagonal entry $\mathit{L}_{ui}=\frac{1}{\sqrt{degree_u \ \cdot \ degree_i}}$. Thus, the weight of node centrality can be mitigated in LightGCN.
>
> From the experiments in the paper, we can observe that simply mitigating the weight of node centrality (e.g., normalization) cannot address popularity bias very well. We speculate that even if the weights of nodes are normalized, through the multi-layer convolution of GCNs, the information of nodes with rich connections can be easily passed to a broad range of nodes, compared to nodes with few connections.
>
> ---
>
> >**Q2**:
> To what extent Proposition 1 is a real new contribution (that can be used for other research, and in what context), or if it's a rather straightforward derivation.
>
> **A2**:
> Proposition 1 is indeed an extension of the classical information bottleneck theory. However, it is specific to our paper that it shows that our Eq. 13 has the noise-invariance property, thus providing theoretical support for our claim to mitigate interaction noises in addition to the experimental results. We would like to clarify that the contribution of this article is the overall methodology. CGI first proposes a multi-view graph representation learning schema constrained by Information Bottlenecks (IB). With learnable graph augmentation and IB, we help representations of different views to capture collaborative information of different semantics that complement each other. In this way, we can learn a "residual" that is difficult to learn from the original graph structure for better mitigating popularity bias and increasing robustness for noise on graph-based recommendation.
>
> ---
>
> >**Q3**:
> The authors don't point out any possible limitation.
>
> **A3**:
> Thanks for the suggestion. A potential limitation of our work is that our work lacks a discussion on how to design good information bottleneck when node features (e.g. age and brand) are introduced, so that the learned graph structure is consistent with the distribution of node features. This could be a promising direction for future work.

---

### Official Review · Reviewer_UmNd · 2022-07-04

**Rating:** 5
**Confidence:** 4
**Soundness:** 3 good
**Presentation:** 3 good
**Contribution:** 3 good

**Summary:**

This paper proposes CGI, a contrastive model for item-user bipartite graphs in recommendation systems. The CGI consists of two parts: learnable graph structure augmentation and information bottleneck losses. Each of these modules reduces the influence of popularity biases and noisy interactions. The experiments on three real-world recommendation tasks demonstrate that CGI outperforms baselines, especially for unpopular items and noisy settings.


**Questions:**

- It is nice to see the statistical significance tests in the result tables. Could you write the standard deviation and the number of actual runs in the experiments?
- How about changing the name from CGI to CGIB? Surveying existing works, models that end with ‘I’ are conventionally InfoMax models (DGI, HDGI, GMI, GCI, …) and models that end with ‘IB’ are usually Information bottleneck models (GIB, HGIB, VIB-GSL, VGIB, …).


**Limitations:**

The authors fairly discussed limitations.


**Strengths And Weaknesses:**

## Strengths

This paper deals with an important problem in recommendation systems. The model design aligns with the motivation, and extensive studies under various scenarios are well-presented. The authors design experiments for sub-problems of this particular domain (popularity biases and interaction noises) and present them in an appropriate format. I like the Effectiveness of Information Bottleneck part (line 340) since this is an essential analysis to support the model design choice of MI minimization rather than maximization.

## Weaknesses

Although I have decided to rate this paper as borderline accept, there are two flaws in this paper. One is a minor problem that can be solved by simple revision, but the other can be critical. Even if a rejection decision is made, I will not object to it.

First, some related works of information bottleneck models on graphs are missing.

- Yu, Junchi, et al. "Graph Information Bottleneck for Subgraph Recognition." International Conference on Learning Representations. 2020.
- You, Yuning, et al. "Bringing your own view: Graph contrastive learning without prefabricated data augmentations." Proceedings of the Fifteenth ACM International Conference on Web Search and Data Mining. 2022.
- Sun, Q., et al. "Graph Structure Learning with Variational Information Bottleneck." Proceedings of the AAAI Conference on Artificial Intelligence. 2022.
- Yu, Junchi, Jie Cao, and Ran He. "Improving subgraph recognition with variational graph information bottleneck." Proceedings of the IEEE/CVF Conference on Computer Vision and Pattern Recognition. 2022. (Yes, I know this paper is very recently published, but it was on arXiv more than three months ago. Moreover, it is deeply related to the authors' work. See the 'Second' section below)

Second, the CGI model might be a straightforward combination of existing approaches but for a specific domain. The technical novelty is somewhat limited. Learnable masking of nodes or edges with reparameterization is a well-known approach for learning graph structures, and using it as augmentations have been proposed recently (as the authors say). The authors use information bottleneck losses in the same form as the original GIB paper (and other variants for graph tasks) with similar motivations. I believe this is an appropriate choice for this problem, but it is difficult to call it novel. In addition, the theoretical result (proposition 1) is a simple application of Proposition 3.1 in [1] and is very similar to Lemma 4.1 in [2].

[1] Achille, Alessandro, and Stefano Soatto. "Emergence of invariance and disentanglement in deep representations." The Journal of Machine Learning Research 19.1 (2018): 1947-1980.

[2] Yu, Junchi, Jie Cao, and Ran He. "Improving subgraph recognition with variational graph information bottleneck." Proceedings of the IEEE/CVF Conference on Computer Vision and Pattern Recognition. 2022.

---

> ### Author Response · Authors · 2022-08-02
> **Response to Reviewer UmNd (Part III)**
>
> >**Q3**:
> Could you write the standard deviation and the number of actual runs in the experiments?
>
> **A3**:
> Each experiment is conducted 5 times to alleviate randomness. We add the corresponding standard deviations in Table 1 as follows:
>
> |          |               |               | yelp2018      |               |
> |----------|---------------|---------------|---------------|---------------|
> |          | NDCG@10       | Recall@10     | NDCG@20       | Recall@20     |
> | BPRMF    | 0.0138±0.0007 | 0.0209±0.0009 | 0.0191±0.0012 | 0.0373±0.0013 |
> | NCF      | 0.0224±0.0011 | 0.0356±0.0019 | 0.0289±0.001  | 0.0566±0.0018 |
> | NGCF     | 0.0242±0.0013 | 0.0384±0.0024 | 0.0319±0.0023 | 0.0629±0.0048 |
> | LightGCN | 0.0344±0.0012 | 0.053±0.0013  | 0.0445±0.0016 | 0.085±0.0022  |
> | DNN+SSL  | 0.0217±0.0006 | 0.0344±0.002  | 0.0286±0.0012 | 0.0564±0.0013 |
> | SGL      | 0.0367±0.0008 | 0.0552±0.0012 | 0.0473±0.0010 | 0.0891±0.0012 |
> | MGIB     | 0.0392±0.001 | 0.0584±0.0016 | 0.0501±0.0008 | 0.0932±0.0012 |
>
> |          |               |               | movielens     |               |
> |----------|---------------|---------------|---------------|---------------|
> |          | NDCG@10       | Recall@10     | NDCG@20       | Recall@20     |
> | BPRMF    | 0.1225±0.0066 | 0.1376±0.0104 | 0.1407±0.0055 | 0.1882±0.0057 |
> | NCF      | 0.143±0.009   | 0.1546±0.0072 | 0.1576±0.0098 | 0.2027±0.005  |
> | NGCF     | 0.1462±0.0139 | 0.1651±0.0154 | 0.1667±0.0068 | 0.2285±0.0114 |
> | LightGCN | 0.1696±0.0099 | 0.1865±0.0086 | 0.1863±0.0035 | 0.242±0.0065  |
> | DNN+SSL  | 0.1096±0.0071 | 0.1238±0.0057 | 0.125±0.0041  | 0.1714±0.0098 |
> | SGL      | 0.18±0.0078   | 0.1965±0.0081 | 0.1972±0.0041 | 0.2518±0.0045 |
> | MGIB     | 0.1979±0.0039 | 0.218±0.0045   | 0.2152±0.0068 | 0.2772±0.0068 |
>
> |          |               |               | douban        |               |
> |----------|---------------|---------------|---------------|---------------|
> |          | NDCG@10       | Recall@10     | NDCG@20       | Recall@20     |
> | BPRMF    | 0.0496±0.001  | 0.0526±0.0006 | 0.0516±0.0007 | 0.0613±0.0015 |
> | NCF      | 0.0694±0.0013 | 0.0706±0.0009 | 0.0659±0.0014 | 0.0734±0.0015 |
> | NGCF     | 0.0794±0.0011 | 0.0823±0.0015 | 0.0784±0.0016 | 0.0897±0.002  |
> | LightGCN | 0.0862±0.0021 | 0.0876±0.0013 | 0.0845±0.0021 | 0.094±0.0019  |
> | DNN+SSL  | 0.0712±0.0015 | 0.0738±0.0014 | 0.0703±0.0022 | 0.0804±0.0022 |
> | SGL      | 0.0912±0.0022 | 0.0906±0.0028 | 0.091±0.0024  | 0.1012±0.0024 |
> | MGIB     | 0.0991±0.0038 | 0.1007±0.0059 | 0.0979±0.0009  | 0.1119±0.0032  |
>
>
> ---
>
> >**Q4**:
> How about changing the name from CGI to CGIB?
>
> **A4**:
> Thanks for your nice suggestion! We will change the name to CGIB in the final revision so as to be consistent with the previous works.

---

> > ### Comment · Reviewer_UmNd · 2022-08-05
> > **Thank you for your detailed responses.**
> >
> > Thank you for your detailed responses.
> >
> > I have read the authors' responses and these resolve my concerns.
> > Looking forward to seeing the final revision reflecting the rebuttal.

---

> > > ### Author Response · Authors · 2022-08-07
> > > **Response on feedback**
> > >
> > > Dear Reviewer UmNd:
> > >
> > > Thank you very much for reading our response, and we are delighted that our contributions were recognized. We will ensure that rebuttal content is included in the updated version based on your suggestions. Thanks again for your valuable comments and suggestions!
> > >
> > > Sincerely thanks,
> > > Authors

---

> ### Author Response · Authors · 2022-08-02
> **Response to Reviewer UmNd (Part II)**
>
> >**Q2**:
> The technical novelty is somewhat limited.
>
> **A2**:
> Our novelty lies in the design of the overall framework, which is a multi-view graph representation learning schema constrained by Information Bottlenecks. First, we agree with the reviewer that learnable masking of nodes or edges is an efficient way to learn graph structures and IB is also a well-tested information-theoretic principle used to find the best tradeoff between accuracy and complexity. By using the IB principle, there are many different implementations, such as works mentioned by the reviewer, which are also compared with our method in response **A1**.
>
> In CGI, we implement the IB principle by contrastive learning of two complete bipartite graphs, so as to guide the learning of the augmentation graph structures, making them as different as possible from the original bipartite graph, while ensuring the recommendation performance on both graphs. Different from existing methods, we do not directly replace the original structure with the learned structure to participate in the downstream tasks. Instead, we let the original graph and the augmentation graph as two views of the data and encourage them to capture view-specific information that complements each other. The representations learned from all views are aggregated together into a compact representation for the downstream recommendation. The reason behind is that through IB, we can obtain a debiased and denoised structure, and through such a structure, we learn a "residual" that is difficult to learn from the original graph structure. Integrating such a residual brings information gain, which is different from existing IB methods that only focus on discarding superfluous information. The GIB mentioned by the reviewer is a typical representative of the existing IB methods, which seek to obtain a subgraph retaining the minimum sufficient information for better robustness of the learned representation.
>
> In addition to the above heuristic differences, some other technical differences between CGI and GIB are as follows. In GIB, the IB for the graph structures relies on some oversimplified priors, e.g., uniform distribution for GIB-Cat and Bernoulli distribution with hyperparameter $\alpha$ for GIB-Bern. And the IB for node representation is to compute a Gaussian distribution that is as different as possible from the mixture of Gaussians learned from the original representation. However, the distribution of node representations may not follow Gaussian distribution and the component number of the mixture of Gaussians is hard to determine. While CGI works with arbitrary distributions and with no hyperparameter that is hard to tune. In addition, GIB is only studied for node classification, which is very different from our collaborative filtering recommendation task.
>
> Interaction noises on the bipartite graph have always been an important problem in graph-based recommender systems.
> The theoretical result (proposition 1) is an extension of the classical information bottleneck theory.
> However, it is specific to our paper that it proves that our objective in Eq. 13 has the noise-invariance property, thus providing theoretical support for our claim to mitigate interaction noises in addition to the experimental proof.

---

> ### Author Response · Authors · 2022-08-02
> **Response to Reviewer UmNd (Part I)**
>
> Many thanks for your ideas and comments. Please find our thoughts about them below.
>
> >**Q1**:
> Some related works of information bottleneck models on graphs are missing.
>
> **A1**:
> Thanks for the comment. As suggested, we will add a detailed discussion on these related works in our revision.
>
> *[2] proposes a variational graph auto-encoder to generate contrastive views and the downstream contrastive learning utilizes IB performing on graph representations as the unsupervised loss. Both [1] and [4] aim to directly reveal the vital substructure in the subgraph level, among which [1] learns a node assignment matrix to extract the subgraph, and implements the IB of two graphs by estimating the KL-divergence from graph latent representation with a statistic network (DONSKER-VARADHAN Representation of KL-divergence). And [4] employs noise injection to manipulate the graph, and customizes the Gaussian prior for each input graph and the injected noise, so as to implement the IB of two graphs with a tractable variational upper bound. Our CGI differs from them, since we do not directly aim to find an optimal graph structure, instead we try to learn the graph structure complementing the original one. Then by integrating different views into a compact representation, we obtain the optimal node representation for the downstream task. [3] learns to mask node feature and generates new structure with the masked feature. Afterward, [3] adopt GNN to learn the distribution of graph representation and utilize the KL-divergence between the learned distribution and the prior distribution to implement the IB.*
>
> *All these methods aim to find a better structure or representation to replace the original graph for the downstream task, while our CGI follows a multi-view representation learning schema. IB is utilized to minimize the mutual information between the original graph and the generated views while maintaining the downstream recommendation performance of each view. Besides the noise-invariance property, IB helps representations of different views to capture collaborative information of different semantics that complement each other.*
>
> Refs:
> [1] Yu, Junchi, et al. "Graph Information Bottleneck for Subgraph Recognition." International Conference on Learning Representations. 2020.
> [2] You, Yuning, et al. "Bringing your own view: Graph contrastive learning without prefabricated data augmentations." Proceedings of the Fifteenth ACM International Conference on Web Search and Data Mining. 2022.
> [3] Sun, Q., et al. "Graph Structure Learning with Variational Information Bottleneck." Proceedings of the AAAI Conference on Artificial Intelligence. 2022.
> [4] Yu, Junchi, Jie Cao, and Ran He. "Improving subgraph recognition with variational graph information bottleneck." Proceedings of the IEEE/CVF Conference on Computer Vision and Pattern Recognition. 2022.

---

### Official Review · Reviewer_5uX4 · 2022-07-09

**Rating:** 6
**Confidence:** 3
**Soundness:** 3 good
**Presentation:** 3 good
**Contribution:** 3 good

**Summary:**

The paper proposes a graph neural network model based on LightGCN using learned parameters to adaptively dropout nodes and edges to obtain multi-view representation and contrastive learning with information bottleneck. Experiments on public datasets show improvements compared to existing models. Ablation studies show effectiveness of different components in the model.

**Questions:**

The main GNN component is based on LightGCN, have you tried with other GNN variants? Do they work well?

From the details of the dataset, it seems that the paper only considers a binary case for Movielens and Douban (1 vs 0) and this seems to simplify the problem. For real recommendation systems, there's a different between recommending an item that user would rate 5 vs 4 (assuming 5 is the highest rating). It would be worth trying to fully utilize the ratings information in these datasets.

If the proposed model aims to address the popularity bias issue, why are items with fewer interactions removed (Yelp with <10, Movielens with <3, Douban with <5)? These are real data and should be considered in the model training and evaluation unless they are corrupt or spammy data points.

**Limitations:**

The model assumes that the data is relatively clean (without much spam or misleading content/item). Real recommendation systems would need to consider filtering out spammy content, misleading and low quality content.

**Strengths And Weaknesses:**

Strength
- The paper explains the motivation behind different model components, e.g., adaptive dropping of edges and nodes to reduce popularity bias and contrastive learning via information bottleneck to reduce noise.
- The paper conducts thorough experiments using 3 public datasets and compared with multiple models. There are also ablation studies to verify different hypothesis in the model. The paper also provides detailed hyperparameters and time complexity analysis. Code was also provided for reproducing the results.

Weakness
- While the paper provides empirical results showing that the model can reduce popularity bias and provide robustness to noise, there are no clear theoretical link provided.

---

> ### Author Response · Authors · 2022-08-02
> **Response to Reviewer 5uX4 (Part II)**
>
> >**Q3**:
> It would be worth trying to fully utilize the rating information in these datasets.
>
> **A3**:
> Thanks for the suggestion. We believe that the introduction of rating information to differentiate the edges on the graph can effectively improve the performance of the recommendation model. However, most existing works on graph-based recommender systems focus on implicit feedback, e.g. [8, 30, 32, 38] in our paper. In this paper, we propose a model-agnostic framework to enhance graph-based recommender systems. Therefore, our problem definition is also consistent with these graph-based recommender systems, so as to better demonstrate the improvement brought by CGI. In practical scenarios, most feedback that a user provides is implicit (e.g., clicks, views), instead of explicit (e.g., ratings, likes/dislikes). As such, implicit feedback is more pervasive in practice but also easily biased and inherently noisy. Such a setting can better demonstrate the ability of our CGI to alleviate the popularity bias and the interaction noises, and also verify the broad adaptability of CGI.
>
> As a model-agnostic framework, CGI also supports introducing rating information by using a specific backbone model. In fact, there are many existing GNNs that can handle different edge weights on the graph, e.g., EGNN [1]. We will explore fully utilizing the rating information in future work.
>
> Refs:
> [1] Gong L, Cheng Q. Exploiting edge features for graph neural networks. 2019. In *Proc. of the IEEE/CVF conference on computer vision and pattern recognition (CVPR)*.
>
> ---
> >**Q4**:
> Why are items with fewer interactions removed?
>
> **A4**:
> We exclude a small portion of items and users that have extremely low interactions in the dataset in order to ensure that our CGI is trained and validated in a high-confidence environment. When the numbers of samples of items and users are too small, the data themselves may not contain enough information that truly reflects the characteristics of users or items. Especially when we further divide them into training and validation sets, the limited useful information may be further split into two parts that do not overlap with each other, making both training and validation lack confidence. The popularity bias in our problem mainly refers to the long-tail distribution of data, that is, the imbalanced interaction numbers between the head and tail users or items, which can bias GCN-based models easily. In this way, when we remove those extreme examples, we can make our verification on the tail data more confident, so as to better demonstrate the ability of CGI to mitigate the popularity bias.

---

> ### Author Response · Authors · 2022-08-02
> **Response to Reviewer 5uX4 (Part I)**
>
> Thanks for your detailed reading and suggestions.
> Please see our detailed response and clarification below:
>
> >**Q1**:
> There are no clear theoretical links provided to show how the model can reduce popularity bias and provide robustness to noise.
>
> **A1**:
> Thanks for the comment. For robustness to noise, we provided theoretical analysis in Section 4.3, which extended the classical information bottleneck theory and proved that our objective in Eq. 13 had the noise-invariance property. For popularity bias, we compare our CGI with those methods creating augmentation by random dropout, e.g., SGL. We believe that random dropout will indiscriminately drop nodes or edges regardless of the corresponding node degrees. Through the message passing mechanism, it is easier for GCNs to reconstruct the missing information of popular users or items, but much harder to reconstruct those isolated nodes with few connections, thus may overemphasize those high-degree nodes when adopting random dropout. In contrast, the proposed learnable graph augmentation strategy can intentionally drop less information of isolated nodes to keep the final recommendation loss to be low. The claim is also supported by an experiment investigating the degree distribution of the dropping nodes in Appendix B.
>
> ---
>
> >**Q2**:
> Do the CGI work well with other GNN variants?
>
> **A2**:
> Thanks for your suggestions. We tried CGI and the baseline SGL on two other popular GNN-based recommenders GC-MC [1] and NGCF [2].
> The experimental results are shown as follows:
>
> |           |                | Yelp2018      |                | Movielens-1M   |               | Douban        |
> |-----------|----------------|---------------|----------------|----------------|---------------|---------------|
> |           | NDCG@10        | Recall@10     | NDCG@10        | Recall@10      | NDCG@10       | Recall@10     |
> | GC-MC     | 0.0214         | 0.0278        | 0.1350         | 0.1491         | 0.0671        | 0.0739        |
> | SGL+GC-MC | 0.0218(+1.9%)  | 0.0281(+1.2%) | 0.1412(+4.6%)  | 0.1577(+5.8%)  | 0.0687(+2.3%) | 0.0762(+3.1%) |
> | CGI+GC-MC | 0.0218(+2.1%)  | 0.0282(+1.7%) | 0.1422(+5.3%)  | 0.1585(+6.3%)  | 0.0687(+2.3%) | 0.0765(+3.5%) |
> |           |                |               |                |                |               |               |
> | NGCF      | 0.0242         | 0.0384        | 0.1462         | 0.1651         | 0.0794        | 0.0823        |
> | SGL+NGCF  | 0.0260(+7.4%)  | 0.0418(+8.9%) | 0.1609(+10.1%) | 0.1871(+13.3%) | 0.0833(+4.9%) | 0.0857(+4.1%) |
> | CGI+NGCF  | 0.0272(+12.5%) | 0.0431(12.1%) | 0.1660(%13.6%) | 0.1937(+17.3%) | 0.0840(+5.7%) | 0.0875(+6.3%) |
>
>
> Both graph contrastive learning methods have shown improvements to the backbones. On NGCF, CGI shows consistent superiority compared to SGL. On GC-MC, CGI does not have significant improvement compared to SGL, probably due to the fact that GC-MC only utilizes one layer of GCN, which means it can only adopt 1-hop neighbors for learning, thus making the learnable augmentation challenging to fetch enough information.
>
> Refs:
> [1]. Rianne van den Berg, Thomas N Kipf, and Max Welling. 2018. Graph Convolutional Matrix Completion. In *Proc. of the 24th ACM International Conference on Knowledge Discovery and Data mining (SIGKDD)*.
> [2]. Xiang Wang, Xiangnan He, Meng Wang, et al. 2019. Neural Graph Collaborative Filtering. In *Proc. of the 42nd International ACM Conference on Research and Development in Information Retrieval (SIGIR)*.

---

> ### Author Response · Authors · 2022-08-08
> **Dear reviewer**
>
> Dear reviewer 5uX4:
>
> Thanks a lot for your efforts in reviewing this paper. We have tried our best to address all your concerns and provided more experiments. Please let us know whether there are any unclear explanations. In addition, if you have any further questions, we will also be very glad to further clarify them.
>
> Sincerely thanks,
>
> Authors

---

### Official Review · Reviewer_cHS4 · 2022-07-12

**Rating:** 6
**Confidence:** 5
**Soundness:** 2 fair
**Presentation:** 3 good
**Contribution:** 2 fair

**Summary:**

This work aims to alleviate the effect of popularity bias and noisy user-item interactions in graph convolutional networks (GCNs) for recommendation. To this end the authors propose a GCN-based recommender method called Contrastive Graph Structure Learning via Information Bottleneck (CGI). In addition to the standard GCN layer producing node representations from the observed user-item graph, CGI includes a node and an edge dropping components whose role is to generate augmentations of the observed graph by dropping some nodes and edges respectively. To improve the quality of the learned augmentations, the authors introduce (in addition to the recommendation objective) an Information Bottleneck (IB)-based contrastive objective, which encourages the representations learned from the augmented graphs to be independent from those learned from the observed graph. The proposed method is evaluated and compared to some existing recommender models on three real-world datasets.

**Questions:**

Please refer to the strengths and weaknesses section.

**Limitations:**

Yes.

**Strengths And Weaknesses:**

Strengths.
1. The paper is well written and technically sound.
2. This work investigates interesting ideas such as the IB-based contrastive objective, as well as learning the graph augmentations instead of relying on random or predefined procedures to generate them.
3. Empirical results show that CGI outperforms the chosen baselines on the recommendation task and seems more robust to noisy user-item interactions.

Weaknesses.
1. The claim regarding the ability of the proposed method to alleviate the popularity bias is not well supported in the paper, nor by theoretical analyses, neither by convincing targeted experiments. For examples, I would recommend reporting statistics about the popularity distribution of the recommended items by the different baselines. Also, some quantitative and qualitative experiments on how popular/rare items are ranked by the different models for a given set of users would be meaningful (see for instance Figures 2 and 3 in [1]).
2. Experiments are weak. Three datasets are considered for evaluation: Yelp, MovieLens and Douban. However, most of the results are on Yelp and/or MovieLens, except in Table 1.  I would recommend reporting the results of every experiment across all the three datasets.

Additional comments/questions.

- In section 4.3, the part related to proposition 1 is a bit hard to follow and connect to the objective of eq. 13. This is due to using different notations for the mutual information terms in the proposition and in eq. 13. Please consider improving the notations.
- I would recommend keeping the legend consistent across all the experiments. For instance, CGI is represented by a green bar in figures 3 and 4, while in figure 5 it is represented by a blue bar.
- What type of significance test is used in the experiments, and how many trials are performed for every algorithm in Table 1.

References.

[1] Liang, Dawen, et al. "Factorization meets the item embedding: Regularizing matrix factorization with item co-occurrence." Proceedings of the 10th ACM conference on recommender systems. 2016.

---

> ### Author Response · Authors · 2022-08-02
> **Response to Reviewer cHS4 (Part II)**
>
> >**Q2**: The results of every experiment across all three datasets should be reported.
>
> **A2**: We repeat every experiment in the paper on all three datasets.
> For experiments in Table 2, we add experiments on **Douban** and show results as follows:
> |              | **NDCG@10** | **Recall@10** |
> |--------------|-------------|---------------|
> | **LightGCN** | 0.0862      | 0.0876        |
> | **CGI**      | _0.0991_    | _0.1007_      |
> | **SGL-ND**   | 0.0835      | 0.0823        |
> | **CGI-ND**   | 0.0903      | 0.0895        |
> | **SGL-ED**   | 0.0912      | 0.0906        |
> | **CGI-ED**   | 0.0965      | 0.0982        |
>
> For other experiments in Figures 3, 4, and 5, we also conduct experiments on every dataset and show the result Figures in the following link: [https://drive.google.com/drive/folders/1gtDCHdhlb_Lp2QHVLZFm9M43awcvCpjs?usp=sharing](https://drive.google.com/drive/folders/1gtDCHdhlb_Lp2QHVLZFm9M43awcvCpjs?usp=sharing). *bias_douban.png* is the result for Douban complementing Figure 3, *noise_douban.png* is the result for Douban complementing Figure 4, and *IB_movielens.png* as well as *IB_douban.png* are results complementing Figure 5. These newly supplemented experiments show great consistency with existing experimental results in the paper, which further supports the conclusions we made in the article.
>
> ---
> >**Q3**: In section 4.3, the part related to proposition 1 is a bit hard to follow and connect to the objective of Eq. 13. Please consider improving the notations.
>
> **A3**:
> Thanks for the comment. Proposition 1 claims $I(\mathcal{G}';\tilde{\mathcal{G}}) \le I(\mathcal{G}; \tilde{\mathcal{G}}) - I(Y_{Rec};\tilde{\mathcal{G}})$, where $\tilde{\mathcal{G}}$ is the learned augmentation graph, $Y_{Rec}$ is the downstream recommendation information, and $\mathcal{G}'$ is the noisy graph structure. For Eq. 13, $\mathbf{E}$ is the node representation from the original graph $\mathcal{G}$, and $\tilde{\mathbf{E}}$ is the node representation from the learned graph $\tilde{\mathcal{G}}$.
>
> Thus minimizing $I(\mathbf{E}; \tilde{\mathbf{E}})$ is to minimize $I(\mathcal{G}; \tilde{\mathcal{G}})$, and minimizing the recommendation loss in Eq.13 is equivalent to maximize $I(Y_{Rec};\tilde{\mathcal{G}})$. So optimizing the IB contrastive objective in Eq. 13 can reduce $I(\mathcal{G}; \tilde{\mathcal{G}}) - I(Y_{Rec};\tilde{\mathcal{G}})$ in Proposition 1, thus can reduce the mutual information between the learned augmentation view and noisy structure $I(\mathcal{G}';\tilde{\mathcal{G}})$.
>
> Following the comment, we will further improve the notations and descriptions in the final version.
>
> ---
>
> >**Q4**: The legend should be kept consistent across all the experiments.
>
> **A4**:
> Thanks for your suggestions. We have re-design Figure 5 as in the following link [https://drive.google.com/file/d/11FYWLznO8IxS_w8VIzUHUw-OQVT7BzGZ/view?usp=sharing](https://drive.google.com/file/d/11FYWLznO8IxS_w8VIzUHUw-OQVT7BzGZ/view?usp=sharing) and will proofread the entire paper to keep the legend consistent across all the experiments in the final version.
>
> ---
>
> >**Q5**: What type of significance test is used in the experiments, and how many trials are performed for every algorithm in Table 1?
>
>
> **A5**:
> In Table 1, each experiment is conducted 5 times to alleviate randomness and we use **two-sample t-test** to indicate the improvements of CGI over the best baseline are statistically significant.

---

> ### Author Response · Authors · 2022-08-02
> **Response to Reviewer cHS4 (Part I)**
>
> Thanks for the comments. In the following, we provide a point-by-point response.
>
>
> >**Q1**: The ability to alleviate the popularity bias is not well supported.
>
> **A1**: In Figure 3, we decompose the RECALL@10 metric by the item popularity in the datasets and report the performance in each item group. The results show that CGI can significantly improve the recommendation accuracy on long-tailed items.
>
> To better support our claim, we conduct more experiments following the suggestion. Specifically, we divide the items into the top 20% head items and the last 80% tail items based on their popularity, as they follow the long tail distribution. We report statistics about the head items and tail items in the top-10 recommendation lists of all users by different baselines as follows:
> |              | **Yelp2018** |        | **Movielens-1M** |        | **Douban** |        |
> |--------------|-------------:|-------:|-----------------:|-------:|-----------:|-------:|
> |              |    _Head_    | _Tail_ |      _Head_      | _Tail_ |   _Head_   | _Tail_ |
> | **LightGCN** | 97.19%       | 2.81%  | 96.11%           | 3.89%  | 97.41%     | 2.59%  |
> | **SGL**      | 93.41%       | 6.59%  | 94.89%           | 5.11%  | 94.97%     | 5.03%  |
> | **CGI**      | 92.72%       | 7.28%  | 93.79%           | 6.21%  | 93.83%     | 6.17%  |
>
> From these statistics, we can observe that tail items have more chance to be recommended by CGI than in other baselines, which verifies the ability of CGI to alleviate the popularity bias. In addition, we have conducted quantitative and qualitative experiments on how rare items are ranked by the different models for a given set of users following [1]. In the Movielens-1M, we randomly select two users and rank all movies for the users with different models. We divide the ranking results into ten groups and show the numbers of rare movies (with less than 20 watches) in each ranking group. (like Figure 2 in [1]). The results are presented in Figure *rank_stat.png* from the following link: [https://drive.google.com/file/d/1tm2BR4SDkX5Bt9Vpadd_Tn9J0UyKGvZp/view?usp=sharing](https://drive.google.com/file/d/1tm2BR4SDkX5Bt9Vpadd_Tn9J0UyKGvZp/view?usp=sharing), where each subfigure represents one user. From the histograms, we can observe that CGI is capable of pushing rare items to the top ranks of the recommendation.
>
> Also, we compare the recommendation results of SGL and our CGI for a particular user (like Figure 3 in [1]), which are as follows:
>
> | **User's watch history**          | **Top recommendation by SGL** | **Top recommendation by CGI** |
> |-----------------------------------|-------------------------------|-------------------------------|
> | Saving Private Ryan (2217)        | Star Wars: Episode IV (2562)  | Pulp Fiction (1709)           |
> | Mission: Impossible (740)         | Matrix (2123)                 | Star Wars: Episode V (2467)   |
> | From Dusk Till Dawn (376)         | Silence of the Lambs (2193)   | Star Wars: Episode IV (2562)  |
> | Braveheart (1927)                 | Fargo (2000)                  | Terminator (539)              |
> | Terminator 2: Judgment Day (1996) | Pulp Fiction (1709)           | Silence of the Lambs (2193)   |
>
> The numbers in parentheses after the movie titles are the numbers of users who have watched this movie in the training set. We find that SGL tends to rank popular movies on the top, while CGI can better balance popularity and relevance. For example, in the top ranks, CGI recommends the "Terminator" which is strongly relevant to "Terminator 2: Judgment Day", even though it's not enough popular.
>
> These new experiments further support our claim, both qualitatively and quantitatively, that CGI can alleviate popularity bias effectively.

---

> ### Author Response · Authors · 2022-08-08
> **Dear reviewer**
>
> Dear reviewer cHS4:
>
> Thanks a lot for your efforts in reviewing this paper. We have tried our best to address all your concerns and provided experiments across all the three datasets. Please let us know whether there are any unclear explanations. In addition, if you have any further questions, we will also be very glad to further clarify them.
>
> Sincerely thanks,
>
> Authors

---

### Meta-Review · Area_Chair_mR4G · 2022-08-27

**Recommendation:** Accept
**Confidence:** Certain

**Metareview:**

All reviews were on the side of acceptance, but opinion varied from borderline to strong acceptance.

Several aspects of the paper were appreciated:
- The problem was considered important, and the model and its technical ideas were considered interesting. - - The model and its components were considered clearly motivated.
- The performance and robustness advantage over baselines in empirical results was appreciated, as were the ablation studies, and the analysis of the information bottleneck part.
- The paper was considered well written and technically sound, and code was provided.

The reviewers dis have several concerns:
- A claim about alleviating popularity bias was considered not well supported. One reviewer considered it was not well supported either by theory or experiments, whereas another reviewer felt that empirical results did show reduction of popularity bias and noise robustness but no clear theoretical link was provided.
- Opinion on experiments varied: three reviewers considered the experiments extensive / thorough / having good results, but one considered the experiments weak and not giving results on all three data sets in each experiment; additional information was provided by authors in a response.
- Some related work on information bottleneck was missing.
- The novelty was considered limited as a combination of existing approaches.

The discussion with the authors resolved at least in part the latter two concerns.

Overall, it seems the paper can be publishable in NeurIPS provided the improvements provided during the discussion phase are sufficiently taken into account in the final manuscript.

**Award:**

No

---

### Decision · Program_Chairs · 2022-09-14

Accept